# HIGH-EFFICIENT DIFFUSION MODEL FINE-TUNING WITH PROGRESSIVE SPARSE LOW-RANK ADAPTATION

**Teng Hu**[1*], **Jiangning Zhang**[2*], **Ran Yi**[1†], **Hongrui Huang**[1], **Yabiao Wang**[2], **Lizhuang Ma**[1]

[1] Shanghai Jiao Tong University      [2] Youtu Lab, Tencent

{hu-teng,ranyi,lzma}@sjtu.edu.cn  2022212016@stu.hit.edu.cn
{vtzhang,caseywang}@tencent.com

## ABSTRACT

The development of diffusion models has led to significant progress in image and video generation tasks, with pre-trained models like the Stable Diffusion series playing a crucial role. However, a key challenge remains in downstream task applications: how to effectively and efficiently adapt pre-trained diffusion models to new tasks. Inspired by model pruning which lightens large pre-trained models by removing unimportant parameters, we propose **SaRA**, a novel model fine-tuning method with progressive **Sp**arse low-**R**ank **A**daptation to make full use of these ineffective parameters and enable the pre-trained model with new task-specified capabilities. In this work, we first investigate the importance of parameters in pre-trained diffusion models and discover that parameters with the smallest absolute values do not contribute to the generation process due to training instabilities. Based on this observation, we propose a fine-tuning method termed SaRA that re-utilizes these temporarily ineffective parameters, equating to optimizing a sparse weight matrix to learn the task-specific knowledge. To mitigate potential overfitting, we propose a nuclear-norm-based low-rank sparse training scheme for efficient fine-tuning. Furthermore, we design a new progressive parameter adjustment strategy to make full use of the finetuned parameters. Finally, we propose a novel unstructural backpropagation strategy, which significantly reduces memory costs during fine-tuning. Our method enhances the generative capabilities of pre-trained models in downstream applications and outperforms existing fine-tuning methods in maintaining model's generalization ability. Source code is available at https://sjtuplayer.github.io/projects/SaRA.

## 1 INTRODUCTION

In recent years, with the development of diffusion models (Ho et al., 2020; Rombach et al., 2022), tasks such as image generation (Van Le et al., 2023; Zhang et al., 2023a), video generation (Guo et al., 2023; Blattmann et al., 2023), and 3D generation (Poole et al., 2022; Sun et al., 2023) have made significant advancements. Pre-trained diffusion models, particularly the Stable Diffusion series (Rombach et al., 2022), have played a crucial role in these developments, including image customization (Van Le et al., 2023), image editing (Kawar et al., 2023), and controllable generation (Zhang et al., 2023a; Mou et al., 2024). Additionally, by leveraging prior information from the image domain, diffusion models have been extended to tasks such as video (Guo et al., 2023; Blattmann et al., 2023) and 3D generation (Poole et al., 2022; Sun et al., 2023). As these applications continue to evolve, a core issue emerges: how to effectively and efficiently fine-tune the foundational pre-trained diffusion models and apply them to new tasks.

Existing fine-tuning methods (Han et al., 2024; Pan et al., 2024; Ansell et al., 2024; Sung et al., 2021; Fang et al., 2024) can be categorized into three categories (Fig. 1): *1)* **Additive fine-tuning (AFT) methods** (Chen et al., 2022), which introduce additional modules to fine-tune the model,

---

*Equal Contribution
†Corresponding Author

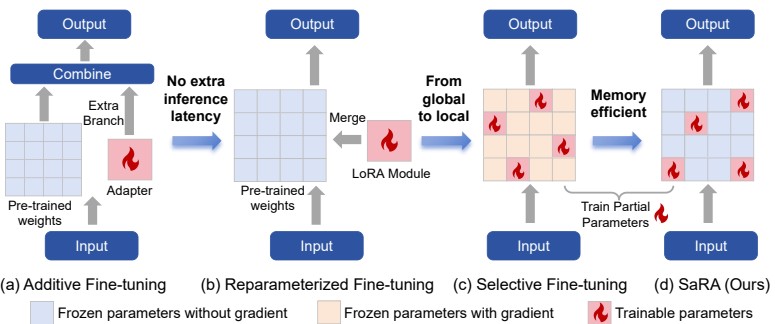

Figure 1: The reparameterized fine-tuning methods (b) address the additional inference latency introduced by additive fine-tuning methods (a) through reparameterizing the pre-trained weights from a global view. Selective fine-tuning methods (c) improve upon global parameter updates by employing sparse updates, which better preserve the model prior by freezing most of the pre-trained parameters. Our SaRA (d) further enhances (c) by significantly reducing memory costs and achieving superior performance in both adaptation capability and prior preservation.

such as adapter-based tuning (Ye et al., 2023; Mou et al., 2024). However, these methods require additional modules and parameters, which has changed the source model, and also introduced additional latency during the inference stage. **2) Reparameterized fine-tuning (RFT) methods** (Hu et al., 2021; Zhang et al., 2023b), which primarily utilize low-rank matrices to learn new information and merge the learned parameters with the pre-trained one, but it still suffers from the risk of overfitting, since all parameters are adjusted by the low-rank matrices globally. Moreover, the choice of rank and the specific layers to which LoRA is applied requires a tailored design for each model. **3) Selective-based fine-tuning (SFT) methods** (Guo et al., 2020; Ansell et al., 2021), which select a subset of the model's existing parameters for fine-tuning. However, the complex parameter selection process and high memory cost restrict their application in diffusion models. Overall, both AFT and RFT methods require model-specific designs, *e.g.,* exploration of which layers to apply Adapters or LoRAs within the model, and the hidden dimension or rank needs to be adjusted according to the specific tasks. The SFT method introduces considerable latency, suffers from hyperparameter sensitivity in parameter selection, and also performs poorly in terms of effectiveness and training efficiency. Therefore, a pressing question arises: Can we design a universal method that is *model-agnostic, does not require hyperparameter searching, inherently avoids overfitting, and simultaneously achieves high-efficiency plug-and-play model fine-tuning*?

Inspired by a theory in model pruning, which posits that within a trained model, there exist parameters with relatively small absolute values that have negligible impact on the model's output, an intuitive idea is: whether we can find a way to leverage these ineffective parameters to make them effective again, and enhance the model's generative capabilities. To achieve this goal, the target "ineffective" parameters we seek must possess two properties: 1) **temporary ineffectiveness**: the parameters themselves have minimal impact on the current model's output; 2) **potential effectiveness**: the parameters are not redundant due to the model structure, but have a certain ability to learn new knowledge (if handled properly, they can be effective again). We first conducted an analysis on the influence of small parameters in pre-trained diffusion models on the model outputs, and found that the smallest 10% (even 20%) of parameters by absolute values did not contribute much to the generative process (Fig. 2, see Sec. 3.1). Furthermore, we examined the potential effectiveness of these parameters and discovered that their ineffectiveness is not inherent (extrinsic) to the model's nature, but rather due to the instability of the training process (see Sec. 3.2). Specifically, the randomness in the training process causes some parameters to approach zero by the end of training. This observation inspired us to rationally utilize these temporally ineffective parameters to make them effective again and fine-tune pre-trained generative models.

Therefore, we propose **SaRA**, a novel fine-tuning method for pre-trained diffusion models that trains **the parameters with relatively small absolute values**. We first identify the "temporally ineffective, potentially effective" parameters as paramters smaller than a threshold in the pre-trained weights. We then efficiently fine-tune these parameters in the pre-trained weights by sparse matrices while preserving prior knowledge. To mitigate the risk of overfitting due to the potential high rank of sparse matrices, we propose a **low-rank sparse training** scheme, which employs a nuclear norm-based low-rank loss to constrain the rank of the learned sparse matrices, achieving efficient fine-tuning of diffusion models. In addition, recognizing that some parameters may not be fully utilized

during the fine-tuning process, we propose a **progressive parameter adjustment strategy**, which introduces a second stage to reselect parameters below the pre-defined threshold and train them, ensuring that almost all parameters contribute effectively. Finally, different from the typical selective PEFT methods that retain the gradient of the entire parameter matrices and require high memory cost (the same as full-parameter fine-tuning), we propose an **unstructural backpropagation strategy** with smaller memory cost. In this strategy, we only retain the gradients for the parameters to be updated, and automatically discard the gradients for other parameters during the backpropagation process. This results in a memory-efficient selective PEFT method, which also advances the development of future selective PEFT techniques. Compared to previous fine-tuning methods (Hu et al., 2021; Valipour et al., 2022; Hayou et al., 2024), our SaRA is capable of effectively enhancing the generative capabilities of the pre-trained model itself, and it also demonstrates the best ability for model adaptation and prior preservation in different downstream tasks.

Contributions of this paper can be summarized in the following four aspects: *1)*We investigate the importance of the parameters in pre-trained diffusion models, revealing the temporal ineffectiveness and potential effectiveness of the parameters with the smallest absolute weight, which motivates us to make full use of these parameters. *2)* We propose SaRA, a novel efficient fine-tuning method based on progressive sparse low-rank adaptation, enabling the model to learn new knowledge without influencing the original generalization ability. *3)* We propose unstructural backpropagation, which resolves the high memory consumption problem of selective PEFT methods and surpasses LoRA in memory efficiency (save more than 40% GPU memory than LoRA and selective PEFT methods). *3)* We efficiently encapsulated and implemented our method in a single line of code modification, which significantly reduces the coding overhead associated with fine-tuning pre-trained models.

## 2 RELATED WORKS

### 2.1 DIFFUSION MODELS

Diffusion models (Ho et al., 2020; Rombach et al., 2022) have demonstrated significant advantages in image generative tasks. Text-to-image models, represented by Stable Diffusion (Rombach et al., 2022), have diversified into various applications. However, their large parameter sizes somewhat limit the feasibility of full fine-tuning to adapt to specific new tasks. Methods such as ControlNet (Zhang et al., 2023a), T2I-Adapter (Mou et al., 2024), and IP-Adapter (Ye et al., 2023) achieve controlled generation under different conditions by adding external networks to diffusion models. Additionally, models like LoRA (Hu et al., 2021) and DreamBooth (Ruiz et al., 2023) enhance the original diffusion models through fine-tuning, enabling them to generate content in new domains and concepts. Furthermore, some video generation models (Guo et al., 2023; Blattmann et al., 2023) are built on diffusion models to achieve video generations and employ Lora and adapters to accomplish controllable video generations.

### 2.2 PARAMETER-EFFICIENT MODEL FINE-TUNING

**Addictive Parameter Fine-tuning (AFT).** AFT introduces additional modules to the model while keeping the pre-trained backbone fixed. Serial Adapter (Houlsby et al., 2019) enhances the Transformer block by adding new modules after the self-attention layer and FFN layer. AdapterFusion (Pfeiffer et al., 2020) streamlines this by inserting adapter layers only after the FFN layers to boost computational efficiency. Parallel adapters, including Adaptformer (Chen et al., 2022), CoDA (Lei et al., 2023), and KronA (Edalati et al., 2022), reorganize the traditionally sequential adapter layers into a parallel side-network, optimizing both performance and efficiency. To further enhance adapter performance and generalization, multi-task learning strategies like AdaMix (Wang et al., 2022), and Hyperformer (Mahabadi et al., 2021) have also been developed.

**Reparameterized Parameter Fine-tuning (RFT).** An early work (Aghajanyan et al., 2020) has verified the presence of low intrinsic dimensionality in pre-trained models. LoRA (Hu et al., 2021) proposes to use a low-rank matrix to learn new feature representations. To address the issue of selecting the appropriate rank, DyLoRA (Valipour et al., 2022) employs a dynamic and search-free approach to obtain the optimal rank. AdaLoRA (Zhang et al., 2023b) decomposes the trainable low-rank matrix using singular value decomposition (SVD) and implements dynamic rank adjustment

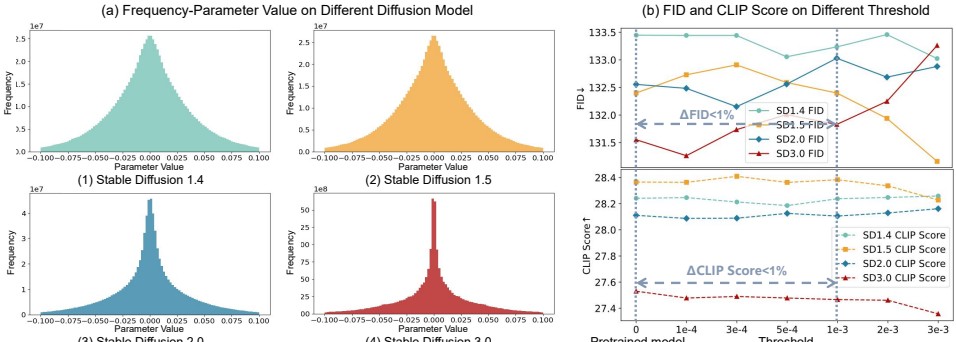

Figure 2: (a) Weight distributions of the pre-trained parameters in Stable Diffusion (SD) 1.4, 1.5, 2.0, and 3.0, which are all similar to a Gaussian distribution, therefore a large number of parameters are around 0. (b) The performance (FID and CLIP Score) of SD Models when the parameters in the pre-trained models with absolute values smaller than a certain threshold are set to 0.

by pruning singular values. Furthermore, numerous subsequent methods (Yang et al., 2023; Ding et al., 2023; Hayou et al., 2024) have aimed to enhance the performance of LoRA.

**Selective Parameter Fine-tuning (SFT).** Selective parameter finetuning (Han et al., 2024) methods finetune a selected subset of the parameters in the pre-trained model. Diffpruning (Guo et al., 2020) fine-tunes specific parameters by learning a mask matrix, constraining its size through a differentiable L0 norm. PaFi (Liao et al., 2023) selects the parameters with the smallest absolute values for learning, while LTSFT (Ansell et al., 2021), grounded in the Lottery Ticket Hypothesis (Frankle & Carbin, 2018), selects the parameters that change the most during fine-tuning. SHiRA (Bhardwaj et al., 2024) proposes a sparse high-rank adaption method to improve the adaptation ability. Essentially, these methods all learn a sparse mask matrix to fine-tune the pre-trained models.

Overall, both AFT and RFT methods require model-specific designs, e.g., determining which layers to apply Adapter or LoRA, and adjusting the hidden dimension or rank according to specific tasks. Additionally, the SFT method introduces significant latency, exhibits hyperparameter sensitivity in parameter selection, and underperforms in terms of effectiveness and training efficiency. In contrast, our SaRA is model-agnostic, which eliminates the need for layer selection, and can effectively finetune the pre-trained model while reducing training costs in both time and memory.

## 3   THE POTENTIAL EFFECTIVENESS OF THE INEFFECTIVE PARAMETERS

### 3.1   INEFFECTIVE PARAMETERS IN STABLE DIFFUSION MODELS

Based on the theorem of model pruning proposed in (Liang et al., 2021), which regards the parameters with the smallest absolute values as "ineffective" parameters, we investigated the effectiveness of these parameters in pre-trained stable diffusion models (version 1.4, 1.5, 2.0, and 3.0). We set parameters with absolute values below a certain threshold $\theta_t$ (from $10^{-3}$ to $10^{-5}$) to zero, and evaluated the performance of the regularized models on generative tasks with CLIP Score (Radford et al., 2021) and Fréchet Inception Distance (FID) (Heusel et al., 2017).

The results are shown in Fig. 2(b). We observed that within a certain threshold range $\theta_t \in (0, 10^{-3}]$, with the small parameters set to 0, the generative ability of the SD models is minimally affected. And in some cases, the regularized model with "ineffective" parameters set to 0 even outperforms the original model (*i.e.,* no parameters are set to 0, with $\theta_t = 0$). Specifically, SD1.4 and SD1.5 show better FID scores than the original model when thresholds are in the range of $\theta_t \in [5 \times 10^{-4}, 10^{-3}]$, and SD2.0 and SD3.0 exhibit superior FID scores at a threshold of $\theta_t = 10^{-4}$. These results show that parameters with the smallest absolute values have a limited impact on the generative process, and in some cases, they may even slightly impair the model's generative ability.

### 3.2   UNSTABLE TRAINING PROCESS CONTRIBUTES TO USELESS PARAMETERS

Sec. 3.1 demonstrated that parameters with smaller absolute values have minimal impact on the generative capability of diffusion models. A natural question arises: **are these currently ineffective**

**parameters caused by the model structure and inherently redundant, or are they caused by the training process and can become effective again?** If it is the former case, the structural design of the model prevents these parameters from learning effective information, then these parameters are redundant and unlikely to be useful in subsequent training processes. While if it is the latter case, these parameters are potentially effective when leveraged rationally in the subsequent training. Therefore, we further investigated the reasons behind the ineffectiveness of these parameters, and found that the ineffectiveness is due to the randomness of the optimization process, rather than an inherent inability caused by model structure.

Specifically, we employed a Stable Diffusion model pre-trained on the FFHQ dataset (Karras et al., 2019), whose parameter matrices are denoted as $P_0$. We recorded the parameters in the pre-trained model with absolute values below the $1\%$ threshold $\theta_t$ by a parameter mask $M$, where $P_M = P_0 \odot M$ denotes the initially below-threshold parameters ($1\%$ of all parameters), and $P_{1-M} = P_0 \odot (1 - M)$ denotes the initially above-threshold parameters ($99\%$ of all parameters), satisfying:

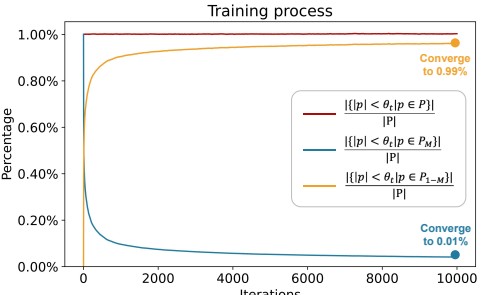

Figure 3: The changes of parameters whose absolute values are bewlow the $1\%$ thesh-old $\theta_t$ during full-parameter fine-tuning. The blue and yellow curves show the proportions of parameters originated from both the initially below-threshold $P_M$ and the initially above-threshold $P_{1-M}$.

$$|p| < \theta_t, \forall p \in P_M,$$
$$|p| \geq \theta_t, \forall p \in P_{1-M}. \tag{1}$$

Then, we continue training this pre-trained model on the FFHQ dataset, and observe the changes of its parameters $P$ during the training process. During this fine-tuning stage, we recorded the "source" of parameters whose absolute values are below the threshold $\theta_t(\{|p| < \theta_t\})$, *i.e.*, whether they are initially below-threshold or initially above-threshold. And we found that these parameters originated from both the initially below-threshold $P_M$ and the initially above-threshold $P_{1-M}$.

The proportions of these two groups and how they change during the finetuning are shown in Fig. 3. As the training progressed, the proportion of $P_M$ remaining below $\theta_t$ gradually decreased from $100\%$ to $1\%$ (blue curve decreases from $1.00\%$ to $0.01\%$); while $1\%$ of the initially above-threshold $P_{1-M}$ eventually fell below $\theta_t$ (yellow curve raises from $0.00\%$ to $0.99\%$). The results indicate that initially ineffective parameters $P_M$ caused by the randomness of the training process, mostly become effective over time (only $1\%$ remaining below threshold). Conversely, some initially effective parameters become ineffective as training continues. This pheonomenon demonstrates that the ineffectiveness of parameters is not inherent to model structure, but rather a result of the stochastic nature of the training process, which causes some parameters to fall below the threshold $\theta_t$ at the last training step coincidentally, making them **temporarily ineffective**. As the training continues, most of these parameters regain effectiveness, proving their **potentially effectiveness,** which motivates us to leverage these temporarily ineffective parameters to fine-tune the pre-trained model.

## 4 PROGRESSIVE SPARSE LOW-RANK MODEL ADAPTATION

Inspired by the potential effectiveness of parameters with the smallest absolute values, as discussed in Sec. 3, we propose SaRA, a novel parameter-efficient fine-tuning method designed to fully utilize these temporarily ineffective parameters. Specifically, we first identify the ineffective parameters in the pre-trained parameters $P_0$ by computing a sparse mask $M = P_0 < \theta_t$, where $\theta_t$ is a threshold and the sparse mask only selects a small portion from all parameters. We then use this sparse mask to update the initially ineffective parameters $P \odot M$, while keeping the initially effective parameters $P \odot (1 - M)$ frozen. This approach enables the pre-trained model to acquire new capabilities for downstream tasks (through the learnable $P \odot M$) while preserving prior information (through the fixed $P \odot (1 - M)$). To avoid the problem of overfitting caused by strong representation ability due to the potential high rank of the learnable sparse matrix $P \odot M$, we propose a nuclear norm-based low-rank loss to mitigate overfitting (Sec. 4.2). In addition, we propose a progressive parameter adjustment strategy to further make full use of the ineffective parameters by progressively reselecting them (Sec. 4.3). Finally, we propose an unstructured backpropagation strategy, which significantly reduces memory costs and can be applied to enhance all selective PEFT methods.

## 4.1 FINE-TUNING ON THE POTENTIAL EFFECTIVE PARAMETERS

In Sec. 3, we have demonstrated that parameters with small absolute values are ineffective in the generative process of diffusion models, and this ineffectiveness is not due to the model's architecture but rather the stochastic nature of the optimization process. Therefore, we propose SaRA, which fine-tunes these temporarily ineffective parameters to adapt the pre-trained diffusion model to downstream tasks, enabling it to learn new knowledge while preserving its original generative capability. Specifically, we first obtain a mask $M$ for the initial parameter set $P_0$, which satisfies:

$$|p| < \theta_t, \forall p \in P_0 \odot M, \tag{2}$$

where $M$ is a **sparse matrix**, since the theshold $\theta_t$ is set low and only selects a small portion from all the parameters. We then use this sparse mask to update the initially ineffective parameters $P_M = P \odot M$, while keeping the initially effective parameters $P \odot (1-M)$ frozen. During training, for the gradient $\nabla P$ of the parameters, we use the pre-defined sparse mask $M$ to retain the gradients we need and update the corresponding parameters $P_M = P \odot M$ by:

$$\nabla P_M = \nabla P \odot M + \mathbf{0} \odot (1 - M), \quad P_{new} = P - \lambda \cdot \nabla P_M. \tag{3}$$

In this way, we can focus on training the ineffective parameters while keeping the other parameters unchanged, ensuring the original generation ability of the pre-trained model is preserved, while learning new knowledge by the parameters $P_M$.

## 4.2 NUCLEAR NORM-BASED LOW-RANK CONSTRAINT

The sparse parameter matrices $P_M$ can sometimes have a high rank, resulting in strong representational capabilities that may lead to overfitting during the training process of downstream tasks. To mitigate this issue, we introduce a nuclear norm-based low-rank constraint on the sparse matrix to prevent the rank from becoming excessively high during the training process.

A direct way to apply low-rank constraint is to minimize the rank of the sparse parameter matrix $Rank(P)$ as a constraint. However, directly minimizing the rank function is computationally intractable due to its non-convex nature. Therefore, we use **nuclear norm** to estimate its rank:

$$\|P_M\|_* = \sum_i \sigma_i(P_M), \text{ where } \sigma_i \text{ are the singular values of } P_M. \tag{4}$$

To compute the nuclear norm $\|P_M\|_*$, we employ the singular value decomposition (SVD) of the matrix $P_M = U\Sigma V^T$, where $U$ and $V$ are orthogonal matrices, and $\Sigma$ is a diagonal matrix containing the singular values $\sigma_i(P_M)$. The subgradient of the nuclear norm at $P_M$ has been derived by (Watson, 1992). Based on this derivation of the nuclear norm gradient, we can ensure that gradient descent methods can be employed to incorporate nuclear norm-based low-rank constraints into the training process, thereby achieving our **nuclear norm-based low-rank constrained loss**:

$$L_{rank} = \|P_M\|_* = \sum_i \sigma_i(P_M). \tag{5}$$

## 4.3 PROGRESSIVE PARAMETER ADJUSTMENT

As discussed in Sec. 3.2 and Fig. 3, when continuing training the pre-trained model, the initially ineffective parameters gradually become above threshold and effective, with only $1\%$ of initially ineffective parameters remaining below threshold eventually. However, the speed at which ineffective parameters become effective (the slope of blue curve in Fig. 3) varies during the finetuning process. In the early stage of the finetuning process (*e.g.,* the first 2.5k iterations), a large portion (over $80\%$) of initially ineffective parameters quickly become effective, with a small part (less than $20\%$) remaining below threshold. However, the speed slows down in the later stage of finetuning: from 2.5k to 8k iterations, the small portion of remaining below-threshold parameters jumps out of the theshold very slowly. However, the finetuning iterations are typically limited (*e.g.,* a few thousands), in which case the slow speed in the later finetuning stage can cause problems: the remaining below-threshold ineffective parameters may not be trained to be effective and fully utilized.

To address this issue, we propose a **progressive parameter adjustment** strategy. To alleviate the slow speed of ineffective parameters becoming effective in the later stage, we reselect the ineffective

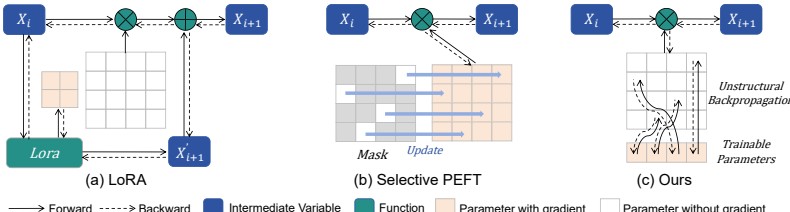

Figure 4: Visualization of our unstructural backpropagation. a) LoRA stores an additional intermediate variable $X'_{i+1}$ in each LoRA layer, and b) selective PEFT methods store the gradients for the whole parameters matrices, causing a waste of memory and computation resources. c) In contrast, our Unstructural Backpropagation method extracts the trainable parameters, sets them as independent leaf nodes, and only retains gradients for them, which largely reduces the memory cost.

parameters that remain below threshold (about $15\%$-$20\%$ of initial ineffective parameters) after the early finetuning stage and focus on optimizing these remaining below-threshold parameters in the subsequent finetuning stage. Compared to finetuning without this reselecting operation, this strategy can quickly make remaining ineffective parameters effective again in the later finetuning stage.

Specifically, we introduce a parameter readjustment phase. After the early finetuning stage (we set the first half of the total iterations as the early finetuning stage, *e.g.,* 2,500 iterations when there are 5,000 finetuning iterations) on the initially selected below-threshold parameters $P^0_{learn}$, we reselect parameters from $P^0_{learn}$ that remain below the predefined threshold as new trainable parameters $P_{learn}$ (which is a subset of $P^0_{learn}$ and typically has $15\%$-$20\%$ of $P^0_{learn}$'s parameters). Then in the subsequent finetuning stage, we only optimize this subset of initial ineffective parameters, and keep other parameters of $P^0_{learn}$ frozen. By focusing on optimizing the small subset of remaining below-threshold parameters, this strategy greatly improves the speed of ineffective parameters jumping out of the threshold in the later finetuning stage, thereby enhancing the model's adaptation capability. In our experiments, we found that under the same number of finetuning iterations, models without the progressive strategy had $15\%$ of $P^0_{learn}$ remained ineffective after finetuning, while models with the progressive strategy only had $2\%$ of $P^0_{learn}$ that were still ineffective. The results indicate this strategy significantly improves the performance of our method during the fine-tuning process.

## 4.4 UNSTRUCTURAL BACKPROPAGATION

Currently, both the LoRA-based methods (the same for the adapter-based methods) and selective PEFT methods cause a heavy burden on the computation resources: 1) For the LoRA-based methods, since the LoRA module is additional to the original model, there is no need to store the gradients of the model parameters, but they still require additional memory costs to store the intermediate variables in the LoRA module, which is shown in Fig. 4 (a). 2) And for the selective PEFT methods, a persistent issue is that they require the same or even more computational resources (especially GPU memory) as full-parameter fine-tuning. Although they only finetune a subset of the model's parameters for fine-tuning, they retain the gradients of the entire parameter matrices $P$, because the mainstream deep learning libraries (such as PyTorch and TensorFlow) only support gradient backpropagation and updates for the entire parameter matrices. Consequently, previous selec-

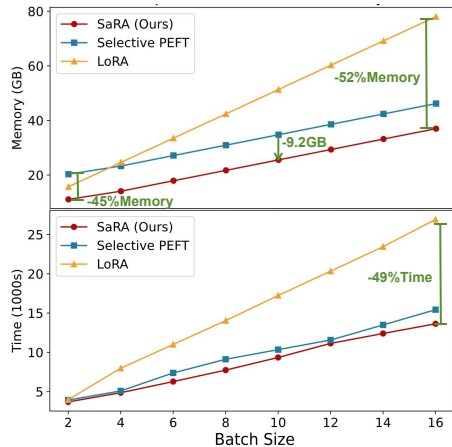

Figure 5: Computation cost on memory and time of different PEFT methods.

tive PEFT methods had to perform gradient backpropagation on the entire parameter matrices $P$, and then use pre-computed mask matrices $M$ to mask out the gradients of unnecessary parameters by $\nabla P_M = M \odot \nabla P$, and perform an overall parameter update by $P_{new} = P - \lambda \nabla P_M$ (visualized in Fig. 4 (b)). This approach necessitates storing the gradients of all model parameters and the additional mask matrices, leading to greater computational resource demands than full-parameter fine-tuning. This clearly contradicts the "efficient" requirements of PEFT and limits the practical applications of such methods.

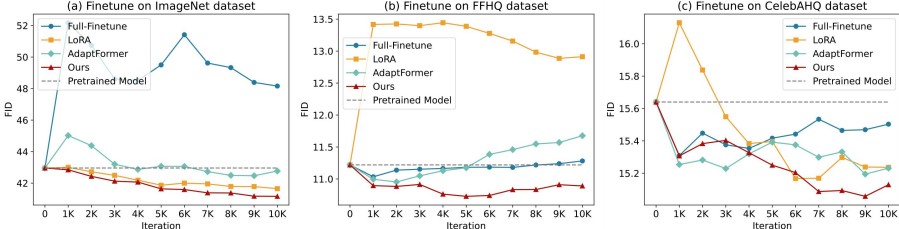

Figure 6: Quantitative comparison among different PEFT methods on Backbone Fine-tuning on ImageNet, FFHQ, and CelebA-HQ datasets. Our method achieves the best FID scores, indicating our method effectively improves the performance of the pre-trained models on the main task.

To address this issue, we propose **Unstructural Backpropagation** (shown in Fig. 4 (c)), which supports efficient gradient backpropagation and updates for unstructured parameters. Different from previous selective PEFT methods that require retaining the gradient for the whole parameter matrices, our Unstructural Backpropagation only needs to retain gradients for the selected subset of below-threshold parameters $P_M$. Specifically, we first store the mask matrices $M$ corresponding to each layer's parameters that need to be trained[1]. In the computational graph, we deviate from the traditional approach of setting model parameters as leaf nodes. Instead, we extract the trainable parameters $P_{learn} = P[M] \in \mathcal{R}^{\|M\|_0}$ and set them as independent leaf nodes, where $[\cdot]$ denotes element-wise indexing of the matrix. I.e., as shown in Fig. 4 (c), we extract the subset of trainable parameters and combine them into a separate parameter vector, and only retain gradients for this vector. Then, during the forward pass, we define an Unstructural Mapping function $UM(\cdot)$ to update the model parameters $P$ by:

$$P = UM(P, P_{learn}, M), where \begin{cases} P[M] & = P_{learn}, \\ P[1 - M] & = P[1 - M]. \end{cases} \tag{6}$$

And the updated model paramaters $P$ will then participate in the training process. During backpropagation, we define the Unstructural Backpropagation function $UB(\cdot)$ to propagate the gradients from the model parameters to the trainable parameters by:

$$\nabla P_{learn} = UB(\nabla P, M) = \nabla P[M]. \tag{7}$$

In this way, during backpropagation, the gradients on the model parameters $\nabla P$ will be automatically cleared, since it is no longer a leaf node, and only the gradients on the learnable parameters $\nabla P_{learn}$ are stored, which significantly reduces the GPU memory during the training process. Notably, unstructual backpropagation is not limited to our method but can be employed in other SFT methods like LT-SFT Ansell et al. (2021), which can advance the development of future SFT fields.

## 5 EXPERIMENTS

To validate the effectiveness of our method, we conduct experiments on various tasks, including backbone fine-tuning, downstream dataset fine-tuning, image customization, and controllable video generation (appendix). We compare our method with three state-of-the-art parameter efficient fine-tunining methods: LoRA (Hu et al., 2021), Adaptformer (Chen et al., 2022), and LT-SFT (Ansell et al., 2021); along with the full-parameter fine-tuning method. We evaluate the generation models by three metrics: *1)* Fréchet Inception Distance (FID) (Heusel et al., 2017), *2)* CLIP Score, and *3)* Visual-Linguistic Harmony Index (VLHI), which balances FID and CLIP Score by:

$$VLHI_i = \frac{\max(\{FID_i\}_{i=1}^n) - FID_i}{\max(\{FID_i\}_{i=1}^n) - \min(\{FID_i\}_{i=1}^n)} + \frac{CLIP_i - \min(\{CLIP_i\}_{i=1}^n)}{\max(\{CLIP_i\}_{i=1}^n) - \min(\{CLIP_i\}_{i=1}^n)}$$

### 5.1 BACKBONE FINE-TUNING

Different from the previous parameter-efficient fine-tuning methods that mainly aim to fine-tune the pre-trained model to downstream tasks, our model enables the pre-trained model to make full use of the parameters. In other words, our finetuning method can improve the performance of pre-trained models on the main task (the original task it is trained on), by optimizing the initially ineffective

---

[1]Since the mask $M$ is of boolean type, it does not consume significant GPU memory.

| Backbone | Params | Model | BarbieCore FID↓ | CLIP↑ | VLHI↑ | Cyberpunk FID↓ | CLIP↑ | VLHI↑ | ElementFire FID↓ | CLIP↑ | VLHI↑ | Expedition FID↓ | CLIP↑ | VLHI↑ | Hornify FID↓ | CLIP↑ | VLHI↑ | Mean FID↓ | CLIP↑ | VLHI↑ |
|---|---|---|---|---|---|---|---|---|---|---|---|---|---|---|---|---|---|---|---|---|
| SD 1.5 | 50M | LoRA | 161.88 | 29.93 | 1.34 | 117.49 | 28.22 | 1.85 | 181.66 | 27.47 | 1.20 | 136.31 | 27.39 | 1.32 | 156.36 | 26.80 | 1.28 | 150.74 | 27.96 | 1.45 |
| | | Adaptformer | 166.09 | 29.00 | 1.00 | 126.21 | 27.13 | 0.66 | 151.27 | 26.57 | 1.29 | 138.01 | 26.41 | 0.63 | 151.53 | 26.20 | 1.18 | 146.62 | 27.06 | 1.18 |
| | | LT-SFT | 157.80 | 23.80 | 0.54 | 123.59 | 25.71 | 0.45 | 171.67 | 25.11 | 0.44 | 139.29 | 27.81 | 1.46 | 158.52 | 26.35 | 1.06 | 150.18 | 25.76 | 0.49 |
| | | SaRA (Ours) | 148.54 | 28.60 | 1.75 | 121.67 | 27.30 | 1.15 | 132.67 | 26.77 | 1.63 | 131.56 | 27.34 | 1.48 | 134.96 | 25.40 | 1.15 | 134.96 | 27.08 | 1.55 |
| | 20M | LoRA | 159.64 | 29.65 | 1.40 | 117.21 | 28.43 | 1.95 | 174.79 | 27.61 | 1.35 | 136.38 | 27.00 | 1.07 | 155.85 | 27.16 | 1.43 | 148.77 | 27.97 | 1.52 |
| | | Adaptformer | 159.02 | 29.08 | 1.34 | 123.88 | 28.07 | 1.19 | 174.17 | 26.53 | 0.95 | 137.03 | 26.67 | 0.83 | 157.09 | 26.63 | 1.20 | 150.24 | 27.39 | 1.21 |
| | | LT-SFT | 156.60 | 23.76 | 0.59 | 119.75 | 25.33 | 0.70 | 191.01 | 25.96 | 0.49 | 144.57 | 28.01 | 1.37 | 165.47 | 26.89 | 1.10 | 155.48 | 25.99 | 0.42 |
| | | SaRA (Ours) | 153.68 | 29.33 | 1.63 | 116.69 | 28.24 | 1.94 | 138.64 | 26.63 | 1.50 | 129.98 | 27.04 | 1.36 | 145.62 | 26.40 | 1.39 | 136.92 | 27.53 | 1.69 |
| | 5M | LoRA | 163.80 | 29.93 | 1.25 | 117.58 | 28.32 | 1.88 | 184.99 | 27.74 | 1.25 | 137.96 | 27.10 | 1.07 | 153.57 | 26.93 | 1.40 | 151.58 | 28.00 | 1.44 |
| | | Adaptformer | 164.22 | 29.37 | 1.14 | 120.98 | 28.11 | 1.48 | 184.84 | 26.66 | 0.84 | 143.01 | 27.35 | 1.01 | 171.34 | 26.85 | 0.94 | 156.88 | 27.67 | 1.13 |
| | | LT-SFT | 169.24 | 24.23 | 0.08 | 127.01 | 25.43 | 0.03 | 202.47 | 26.90 | 0.68 | 153.49 | 27.96 | 0.97 | 176.41 | 27.34 | 1.00 | 165.72 | 26.37 | 0.27 |
| | | SaRA (Ours) | 153.69 | 29.39 | 1.64 | 118.74 | 28.17 | 1.72 | 174.86 | 27.04 | 1.13 | 134.45 | 27.06 | 1.18 | 157.24 | 26.97 | 1.33 | 147.80 | 27.73 | 1.44 |
| | 860M | Full-finetune | 147.81 | 27.77 | 1.65 | 120.22 | 27.84 | 1.47 | 136.49 | 25.10 | 0.95 | 129.07 | 26.75 | 1.21 | 134.86 | 24.64 | 1.00 | 133.69 | 26.42 | 1.30 |
| SD 2.0 | 50M | LoRA | 157.41 | 29.81 | 1.64 | 133.22 | 28.00 | 1.52 | 187.32 | 27.70 | 1.29 | 148.18 | 27.58 | 1.38 | 169.92 | 26.99 | 1.09 | 159.21 | 28.02 | 1.51 |
| | | Adaptformer | 161.87 | 30.78 | 1.75 | 138.02 | 27.85 | 1.12 | 179.44 | 27.35 | 1.26 | 162.45 | 27.06 | 0.47 | 175.39 | 26.59 | 0.76 | 163.43 | 27.93 | 1.25 |
| | | LT-SFT | 164.80 | 28.13 | 0.59 | 134.97 | 26.40 | 0.59 | 183.23 | 25.90 | 0.50 | 153.94 | 27.88 | 1.33 | 167.19 | 26.83 | 1.08 | 160.83 | 27.03 | 0.57 |
| | | SaRA (Ours) | 162.72 | 29.72 | 1.31 | 135.05 | 28.30 | 1.55 | 151.82 | 27.24 | 1.68 | 138.77 | 26.30 | 0.96 | 165.62 | 26.71 | 1.05 | 150.80 | 27.65 | 1.55 |
| | 20M | LoRA | 161.92 | 30.18 | 1.52 | 129.01 | 28.36 | 2.00 | 190.90 | 27.72 | 1.24 | 147.05 | 27.60 | 1.44 | 168.03 | 26.97 | 1.13 | 159.38 | 28.16 | 1.63 |
| | | Adaptformer | 160.29 | 30.42 | 1.70 | 141.80 | 27.92 | 0.89 | 190.57 | 27.33 | 1.05 | 157.31 | 27.07 | 0.69 | 175.39 | 26.59 | 0.76 | 165.07 | 27.86 | 1.13 |
| | | LT-SFT | 168.09 | 28.29 | 0.47 | 135.03 | 26.47 | 0.62 | 194.17 | 26.64 | 0.66 | 155.51 | 27.88 | 1.27 | 174.64 | 27.12 | 1.04 | 165.48 | 27.28 | 0.59 |
| | | SaRA (Ours) | 164.57 | 30.22 | 1.39 | 134.28 | 28.29 | 1.60 | 163.67 | 27.90 | 1.79 | 149.29 | 27.01 | 0.98 | 165.62 | 26.71 | 1.05 | 155.49 | 28.03 | 1.68 |
| | 5M | LoRA | 162.47 | 29.91 | 1.39 | 132.35 | 28.13 | 1.65 | 183.55 | 27.68 | 1.34 | 152.69 | 27.41 | 1.09 | 164.00 | 26.81 | 1.15 | 159.01 | 27.99 | 1.49 |
| | | Adaptformer | 162.25 | 30.52 | 1.63 | 143.41 | 27.69 | 0.66 | 188.42 | 27.45 | 1.15 | 160.23 | 27.37 | 0.76 | 180.07 | 26.72 | 0.71 | 166.88 | 27.95 | 1.12 |
| | | LT-SFT | 175.45 | 28.74 | 0.23 | 137.84 | 26.55 | 0.46 | 209.51 | 27.29 | 0.70 | 161.67 | 27.90 | 1.03 | 186.69 | 27.62 | 1.00 | 174.23 | 27.62 | 0.52 |
| | | SaRA (Ours) | 165.57 | 30.58 | 1.47 | 136.89 | 27.77 | 1.15 | 174.73 | 27.60 | 1.45 | 150.89 | 27.29 | 1.09 | 166.40 | 26.90 | 1.13 | 158.90 | 28.03 | 1.53 |
| | 866M | Full-finetune | 160.87 | 29.30 | 1.25 | 133.19 | 28.33 | 1.70 | 198.45 | 25.81 | 0.19 | 137.84 | 26.74 | 1.27 | 145.99 | 25.64 | 1.00 | 155.27 | 27.16 | 0.93 |
| SD 3.0 | 50M | LoRA | 165.22 | 29.57 | 1.28 | 123.59 | 28.38 | 1.59 | 187.26 | 27.54 | 1.36 | 148.35 | 26.83 | 1.50 | 169.00 | 26.96 | 1.35 | 158.69 | 27.85 | 1.52 |
| | | Adaptformer | 164.09 | 29.81 | 1.43 | 126.73 | 28.23 | 1.38 | 186.05 | 27.14 | 0.93 | 156.77 | 27.12 | 1.62 | 180.11 | 26.91 | 1.09 | 162.75 | 27.84 | 1.41 |
| | | LT-SFT | 209.04 | 29.45 | 0.39 | 158.59 | 27.72 | 0.10 | 208.94 | 27.03 | 0.14 | 204.16 | 26.02 | 0.00 | 189.26 | 26.06 | 0.38 | 194.00 | 27.26 | 0.18 |
| | | SaRA (Ours) | 170.73 | 30.63 | 1.73 | 128.35 | 28.45 | 1.50 | 179.87 | 27.63 | 1.68 | 137.92 | 26.47 | 1.35 | 158.86 | 27.10 | 1.65 | 155.15 | 28.06 | 1.77 |
| | 20M | LoRA | 156.23 | 30.18 | 1.77 | 123.12 | 28.22 | 1.48 | 187.14 | 27.76 | 1.62 | 143.59 | 26.96 | 1.68 | 174.62 | 26.63 | 1.03 | 156.94 | 27.95 | 1.64 |
| | | Adaptformer | 174.32 | 30.30 | 1.49 | 128.73 | 28.31 | 1.39 | 175.60 | 27.77 | 1.96 | 150.69 | 26.50 | 1.19 | 174.61 | 26.56 | 0.99 | 160.79 | 27.89 | 1.50 |
| | | LT-SFT | 167.02 | 29.19 | 1.05 | 154.04 | 27.67 | 0.19 | 203.23 | 26.91 | 0.16 | 155.68 | 26.48 | 1.10 | 177.42 | 26.21 | 0.71 | 171.48 | 27.29 | 0.77 |
| | | SaRA (Ours) | 166.21 | 30.41 | 1.70 | 126.69 | 28.19 | 1.35 | 180.74 | 27.31 | 1.28 | 150.15 | 26.88 | 1.52 | 163.78 | 27.01 | 1.48 | 157.52 | 27.96 | 1.64 |
| | 5M | LoRA | 161.80 | 30.14 | 1.64 | 124.17 | 28.06 | 1.33 | 174.66 | 27.27 | 1.40 | 149.85 | 27.01 | 1.63 | 172.56 | 26.88 | 1.23 | 156.61 | 27.87 | 1.59 |
| | | Adaptformer | 168.98 | 30.50 | 1.69 | 127.35 | 27.89 | 1.11 | 204.69 | 27.71 | 1.05 | 158.60 | 27.03 | 1.52 | 182.22 | 26.88 | 1.03 | 168.37 | 28.00 | 1.40 |
| | | LT-SFT | 158.26 | 29.29 | 1.27 | 134.81 | 27.69 | 0.75 | 181.68 | 27.27 | 1.20 | 153.52 | 27.20 | 1.74 | 193.25 | 26.61 | 0.63 | 164.30 | 27.61 | 1.20 |
| | | SaRA (Ours) | 174.42 | 30.60 | 1.64 | 125.14 | 28.91 | 1.94 | 194.79 | 27.63 | 1.24 | 157.20 | 27.17 | 1.65 | 181.39 | 27.20 | 1.24 | 166.59 | 28.30 | 1.68 |
| | 2085M | Full-finetune | 162.33 | 28.69 | 0.88 | 151.57 | 27.59 | 0.20 | 174.12 | 27.16 | 1.29 | 135.28 | 26.09 | 1.06 | 144.56 | 25.58 | 1.00 | 153.57 | 27.02 | 1.00 |

Table 1: Comparison with different parameter-efficient fine-tuning methods on Stable Diffusion 1.5, 2.0, and 3.0. For most of the conditions, our model achieves the best FID and VLHI score, indicating that our model learns domain-specific knowledge successfully while keeping the prior information well. **Bold** and underline represent the best and second best results, respectively.

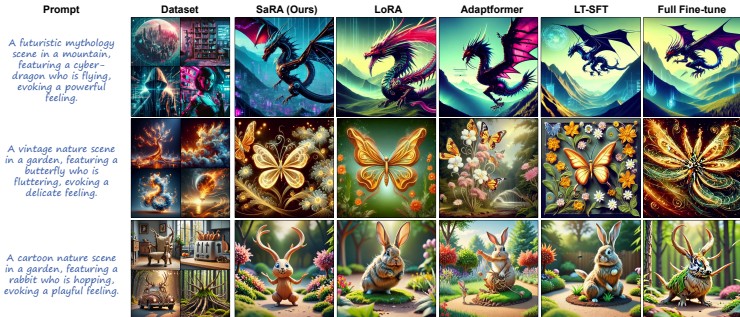

Figure 7: Comparison of the generated images between different PEFT methods.

parameters to be effective and thus increasing the number of effective parameters. Therefore, apart from experimenting on downstream tasks like traditional PEFT methods, we first apply our method to the main task of the pre-trained model, continuing to fine-tune the backbone on the original training dataset, in order to explore whether our method can enhance the base model's performance. Specifically, we employ the pre-trained Stable Diffusion models on ImageNet (Deng et al., 2009), FFHQ (Karras et al., 2019), and CelebA-HQ (Karras et al., 2017) datasets, and fine-tune them on these pre-trained datasets for 10K iterations. We compare our method with full-parameter finetuning, LoRA, AdaptFormer, and LT-SFT by computing the FID metric between 5K generated data and 5K randomly sampled data from the source dataset. The results are shown in Fig. 6, which demonstrates that our method achieves the best FID scores, indicating our method effectively improves the performance of the pre-trained models on the main task.

## 5.2 MODEL FINE-TUNING ON DOWNSTREAM TASKS

**Downstream Dataset Fine-tuning.** In this experiment, we choose 5 widely-used datasets from CIVITAI[2] with 5 different styles to conduct the fine-tuning experiments, which are Barbie Style,

---

[2]https://civitai.com/articles/2138/lora-datasets-training-data-list-civitai-dataset-guide

Cyberpunk Style, Elementfire Style, Expedition Style and Hornify Style. To comprehensively compare PEFT methods, we conduct three sets of experiments for each PEFT method on Stable Diffusion 1.5, 2.0, and 3.0, with selected trainable parameter sizes of 50M, 20M, and 5M. We compute the FID score, CLIP score for the generated data, along with VLHI, which measures both the style (FID) and generalization (CLIP score). The quantitative results are shown in Tab. 1, from which we can draw the following conclusions: *1)* Our model can always achieve the best VLHI on average across five datasets, indicating that our model can preserve the prior information in the pre-trained model well (a good CLIP score), while learning as much task-specific knowledge as possible (a good FID), outperforming all the other PEFT methods and full-finetune method; *2)* As the number of learnable parameter increases, our model can learn more task-specific knowledge (better FID), but may lose part of the prior information (lower CLIP score); *3)* For Stable Diffusions 1.5 and 2.0, our model achieves the best FID and usually the second best CLIP score on average across five datasets, and under different parameter numbers; while for Stable Diffusion 3.0, which has much more parameters than SD 1.5 and 2.0, our model achieves the best CLIP score and usually the second best FID on average across five datasets. The results indicate that for a larger pre-trained model, more learnable parameters are needed to learn the task-specific knowledge well. Moreover, we provide some qualitative comparisons in Fig. 7, which shows the superior generation quality of our method (See appendix for more details.).

## 5.3 ABLATION STUDIES

We conduct ablation studies to validate the effectiveness of our proposed modules: the progressive parameter adjustment (PPA), and the low-rank constrained loss ($\mathcal{L}_{rank}$). Then, we further assess the effectiveness of training parameters with the smallest absolute values, by comparing different parameter-selection strategies, including selecting the largest parameters

| Method | FID ↓ | CLIP Score ↑ | VLHI ↑ |
|---|---|---|---|
| w/o. $PPA$ & $\mathcal{L}_{rank}$ | 134.75 | 27.01 | 1.16 |
| w. $PPA$, w/o. $\mathcal{L}_{rank}$ | 130.95 | 26.66 | 1.56 |
| w. $\mathcal{L}_{rank}$, w/o. $PPA$ | 135.31 | 27.12 | 0.89 |
| **w. PPA & $\mathcal{L}_{rank}$ (Ours)** | 131.56 | **27.34** | **1.79** |
| Tuning Largest Parameters | **130.55** | 25.42 | 1.00 |
| Tuning Random Parameters | 133.57 | 26.58 | 0.97 |

Table 2: Ablation studies on six ablated models.

and random parameters. We conduct downstream dataset finetuning experiments using the Expedition dataset comparing six ablated models: *1)* model without PPA and $\mathcal{L}_{rank}$, *2)* model with PPA but without $\mathcal{L}_{rank}$, *3)* model with $\mathcal{L}_{rank}$ but without PPA, *4)* model with both PPA and $\mathcal{L}_{rank}$ (Ours), *5)* model fine-tuned with the largest absolute values parameters, and *6)* model fine-tuned with randomly selected parameters. The quantitative metric results are presented in Tab. 2: *1)* The model without both the PPA and $\mathcal{L}_{rank}$ results in a poor FID and low CLIP score. *2)* Introducing PPA improves the FID but decreases the CLIP score, indicating its effectiveness in learning task-specific knowledge. *3)* Incorporating $\mathcal{L}_{rank}$ helps achieve a better CLIP score, but results in a worse FID, indicating its effectivenes in better preserving the model prior knowledge, but with a loss of task-specific information. *4)* Regarding parameter-selection strategies, fine-tuning the largest absolute values parameters yields a relatively good FID but the worst CLIP score, suggesting that fine-tuning the most effective parameters severely disrupts the model's prior knowledge and leads to worse content-text consistency. *5)* Moreover, fine-tuning randomly selected parameters results in both poor FID and CLIP scores, indicating randomly selecting parameters to finetune is unable to learn task-specific knowledge and preserve the model's prior. *6)* In contrast, our model achieves the best VLHI, validating its effectiveness in both fitting capability and prior preservation. More analysis of the hyperparameters is presented in the appendix.

## 6 CONCLUSION

In this paper, we propose SaRA, a novel parameter-efficient fine-tuning method, which makes full use of the ineffective parameters with the smallest absolute values in the pre-trained model. We propose a nuclear norm-based low-rank loss to constrain the rank of the learned sparse matrices, thereby avoiding model overfitting. Moreover, we design a progressive parameter adjustment strategy, which can further improve the effectiveness of the fine-tuned parameters. Finally, we propose a novel unstructural backpropagation method, largely saving the memory cost during parameter fine-tuning, which can also reduce the memory costs for other selective PEFT methods. Extensive experiments demonstrate the effectiveness of our method, which achieves the best fitting capability while keeping the prior information of the pre-trained model well.

## ACKNOWLEDGMENTS

This work was supported by National Natural Science Foundation of China (No. 62302297,72192821,62272447,62472282,62472285), Shanghai Sailing Program (22YF1420300), Young Elite Scientists Sponsorship Program by CAST (2022QNRC001), the Fundamental Research Funds for the Central Universities (YG2023QNB17, YG2024QNA44), Beijing Natural Science Foundation (L222117).

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
