# OpenReview forum: "SaRA: High-Efficient Diffusion Model Fine-tuning with Progressive Sparse Low-Rank Adaptation"
_ICLR.cc/2025/Conference — ICLR 2025 Poster_

### Official Review · Reviewer_GTZk · 2024-11-01

**Soundness:** 2
**Presentation:** 3
**Contribution:** 2
**Rating:** 6
**Confidence:** 2

**Summary:**

This paper presents SaRA, a method for fine-tuning pre-trained diffusion models that introduces progressive Sparse low-Rank Adaptation (SaRA) to enhance efficiency and reduce memory costs in adapting diffusion models to new tasks. The proposed method leverages parameters with low absolute values, presumed to have limited initial impact on the model’s performance, making them suitable for fine-tuning. SaRA combines sparse parameter updates with a nuclear norm-based low-rank constraint to mitigate overfitting. It also introduces a progressive parameter adjustment and unstructured backpropagation strategy, aimed at further memory efficiency. Extensive experiments demonstrate SaRA's superiority over traditional fine-tuning methods on image generation and customization tasks.

**Strengths:**

The paper clearly explains the intuition in parameter efficiency by proposing the use of low absolute value parameters for adaptive updates, effectively avoiding overfitting through a nuclear norm constraint.  The progressive parameter adjustment strategy positively contributes to the stability and convergence of model performance, while the unstructured backpropagation method effectively reduces memory costs, making SaRA a practical solution in resource-constrained environments. Additionally, the extensive experiments cover various tasks, such as image generation and customization, thoroughly validating the advantages of the SaRA method in balancing the preservation of model priors and the learning of task-specific knowledge.

**Weaknesses:**

My main concern centers on the assumption underlying this approach—that parameters with the smallest absolute values are inherently ineffective. This seems more empirical than rigorously substantiated. While small absolute values can indeed correlate with lower impact in certain contexts, particularly in pruning, their effectiveness actually depends on the model architecture and specific task. Small value parameters may exert less direct influence on the output, but they are not intrinsically ineffective; their impact can vary depending on training dynamics, model structure, and optimization objectives.

Some minor weaknesses include:
* The generalizability of the 'adapting small-value parameters' strategy across different architectures is crucial to ensure broader applicability. This paper only investigates the phenomenon of the pre-trained Stable diffusion models. In this sense, I'm worrying about whether it can be applied across different network architectures or frameworks.
* In the caption of Figure 3, weight distributions are claimed to be Gaussian without further clarification, which seems empirical rather than solid.
* The choice of threshold for selecting low-absolute-value parameters could be better justified. If this choice is sensitive, it could limit SaRA's robustness and generalizability across different diffusion models and tasks. An ablation study on this threshold choice would strengthen the claims.
* While the experiments show SaRA's success, the paper could benefit from an analysis of its limitations. For instance, discussing cases where SaRA may not perform as well, such as tasks requiring extensive re-training of high-impact parameters, would improve the comprehensiveness of the evaluation.

**Questions:**

See above.

---

> ### Author Response · Authors · 2024-11-23
> **Response to Reviewer GTZk (1/2)**
>
> We sincerely thank you for the valuable suggestions to improve our work. The following content provides a point-by-point response to address your concerns. The corresponding modifications have been refined in the revised submission, highlighted in blue.
>
> **Q1: My main concern centers on the assumption underlying this approach—that parameters with the smallest absolute values are inherently ineffective. This seems more empirical than rigorously substantiated.**
>
> **A1**: Thanks for your question. **1)** Many works[1,2,3,4] have studied the influence of these parameters with the smallest absolute values and confirmed the ineffectiveness of these parameters across many model architectures, including the most widely-used Convolution and Attention modules. Therefore, we further conduct experiments on stable diffusion series (SD 1.5, 2.0, 3.0, XL) and validate the ineffectiveness of these parameters in Sec. 3.1. Although this is not strictly proved, but extensive experiments show that the mainstream models have the same property. Maybe there exist some exceptions, therefore we will include this into the limitation part (Sec. M in Appendix) for a more comprehensive analysis. **2)** Also, some pruning studies [5,6] also find that removing unnecessary weights can reduce the model's complexity and decrease the risk of overfitting. Setting these parameters to zero can suppress noise, allowing the model to focus more on important features, thereby improving its performance.
>
> **Q2: The generalizability of the 'adapting small-value parameters' strategy across different architectures is crucial to ensure broader applicability.**
>
> **A2:** Thanks for your question. **1)** Stable diffusion 1.0 and 3.0 have totally different model architectures, where SD 1.0 employs Unet structure and SD 3.0 employs DiT structure. Our model performs well on both of the two model architectures. **2)** Moreover, we also validate the effectiveness of our model on SD XL, which is a two-stage diffusion model. Our SaRA also achieves the best performance on it, validating the robustness of our model.
>
> **Q3: In the caption of Figure 3, weight distributions are claimed to be Gaussian without further clarification, which seems empirical rather than solid.**
>
> **A3:** Thanks for your question. For a continuous model, due to random data sampling and the stochastic nature of the noise addition and denoising processes in diffusion models, it is reasonable to assume that the probability of any given parameter's gradient update increasing or decreasing is uniform. Assume that each parameter's gradient update $\Delta$ $\theta$ is an independent random variable, and the probability of increasing or decreasing is uniform (i.e., symmetric distribution). We can then assume that the expected value \( $\mu$ \) of these gradient updates is 0, and the variance is \($\sigma^2$ \).
>
> For N gradient updates, the cumulative effect is:
> $$
> \theta = \sum_{i=1}^N \Delta \theta_i
> $$
> According to the Central Limit Theorem, when \( N \) is large, the cumulative gradient update \( $\theta$ \) will approach a normal distribution:
> $$
> \theta \sim \mathcal{N}(0, N\sigma^2)
> $$
> Therefore, for a model with high randomness, the parameters are very likely to be a Gaussian distribution. However, since this is not strict proof, we modify the corresponding Caption from "which all belong to a Gaussian distribution" to "which are all similar to a Gaussian distribution". Thanks for your reminding to improve our work.
>
> **Q4: The choice of threshold for selecting low-absolute-value parameters could be better justified. If this choice is sensitive, it could limit SaRA's robustness and generalizability across different diffusion models and tasks. An ablation study on this threshold choice would strengthen the claims.**
>
> **A4:** Thanks for your suggestion. We have conducted experiments on different thresholds in Tab.1, where the parameter numbers are controlled by the threshold. To study the influence of the threshold more comprehensively, we further conduct an experiment on different thresholds (ranging from 2e-4 to 1e-3) on Expedition dataset and SD 1.5, where the results are shown below:
>
> | Threshold | 2e-4   | 8e-4   | 2e-3   | 5e-3   | 1e-2   | 5e-2   | 1e-1   |
> | :--------- | :------: | :------: | :------: | :------: | :------: | :------: | :------: |
> | FID       | 134.45 | 129.98 | 132.54 | 131.05 | 130.42 | 130.71 | 129.88 |
> | CLIP      | 27.06  | 27.04  | 27.38  | 27.21  | 27.15  | 27.04  | 27.02  |
>
> It can be seen that when the threshold is too small (e.g., 2e-4), the FID becomes much higher, indicating learning less target domain knowledge. And when the threshold is large (i.e., threshold>2e-3), the model performs quite stably. Since we have a low-rank loss, the model with a high threshold also keeps the CLIP score well. In summary, our SaRA performs well in different thresholds, demonstrating the robustness of our model. (supplemented in Sec. H-Tab. 8)

---

> > ### Author Response · Authors · 2024-11-23
> > **Response to Reviewer GTZk (2/2)**
> >
> > **Q5: While the experiments show SaRA's success, the paper could benefit from an analysis of its limitations. For instance, discussing cases where SaRA may not perform as well, such as tasks requiring extensive re-training of high-impact parameters, would improve the comprehensiveness of the evaluation.**
> >
> > **A5:** Thanks for your suggestion. One of the key limitation of SaRA is that, since we aim to fine-tune the ineffective parameters in a pre-trained model, if the model size is not big enough like diffusion model (usually more than 100M parameters), the ineffective parameters may be not enough to fine-tune the model to the downstream dataset. Therefore, our SaRA is more suitable for fine-tuning large models rather than tiny models. Moreover, as illustrated in the first question, there is no strict proof on the ineffectiveness of the parameters with the smallest absolute values, therefore we should be cautious if there are some exceptions, which may lead to a less effective performance of our SaRA. We have included this into the limitation section in Sec. M in Appendix.
> >
> >  **Reference**
> >
> > [1] Liang T, Glossner J, Wang L, et al. Pruning and quantization for deep neural network acceleration: A survey[J]. Neurocomputing, 2021.
> >
> > [2] Lei W, Chen H, Wu Y. Compressing deep convolutional networks using k-means based on weights distribution[C]. Proceedings of the 2nd International Conference on Intelligent Information Processing, 2017.
> >
> > [3] Hao L, Asim K, Igor D, et al. Pruning Filters for Efficient ConvNets[C]. ICLR, 2016.
> >
> > [4] Han S, Pool J, Tran J, et al. Learning both weights and connections for efficient neural network[J]. NeurIPS, 2015.
> >
> > [5] Han S, Pool J, Tran J, et al. Learning both weights and connections for efficient neural network[J]. NeurIPS, 2015.
> >
> > [6] Choudhary T, Mishra V, Goswami A, et al. A comprehensive survey on model compression and acceleration[J]. Artificial Intelligence Review, 2020.

---

> ### Comment · Reviewer_GTZk · 2024-11-24
>
> I appreciate the authors' effort in providing additional results and explanations. Since most of the concerns have been addressed in the rebuttal I'm raising my score.

---

> > ### Author Response · Authors · 2024-11-24
> >
> > Dear Reviewer  GTZk,
> >
> > Thank you for your positive and encouraging feedback on our work! We’re delighted to hear that our rebuttal has addressed your concerns. We sincerely appreciate the time and effort you’ve invested in providing detailed reviews and valuable suggestions to help improve our work.
> >
> > Best regards,
> > The Authors

---

### Official Review · Reviewer_bh95 · 2024-11-01

**Soundness:** 3
**Presentation:** 3
**Contribution:** 3
**Rating:** 6
**Confidence:** 3

**Summary:**

The paper proposes a PEFT method for diffusion models, which progressively trains selectively chosen parameters with small, inefficient values. To prevent memory waste, it avoids storing the gradient of all parameters and instead stores only the chosen parameters in separate nodes, which are then replaced after training.

**Strengths:**

- Unlike methods like LoRA that require additional parameters, this approach selects parameters within the existing model for fine-tuning, minimizing additional memory usage.
- Existing selective PEFT methods continuously update masks, which requires storing gradients for all parameters, making them inefficient. In contrast, this paper’s method progressively trains only the fixed parameters at each stage, storing only certain gradients, making it more memory-efficient.
- The paper conducts extensive comparison experiments with various versions of Stable Diffusion (SD) and different sizes of fine-tuning parameters.

**Weaknesses:**

- Although the paper differentiates itself from selective fine-tuning, I think this method still appears to be a form of selective fine-tuning. The memory efficiency improvement seems to stem from implementing a separate node for gradient storage and selectively fine-tuning only those nodes, rather than from an inherent algorithmic difference.
- Selecting parameters based on a specific threshold seems not a new concept. For example, the related work PaFi is described that it also trains based on absolute values.
- The reasoning behind inefficient parameters becoming efficient during training due to the randomness of the training process is unclear. Since the initial weights are set randomly, many values are likely to change through training. There is no strong basis to assume a correlation with the initial values. Even parameters with initially small values are expected to converge toward the average distribution, making Figure 3 somewhat self-evident.
- Although selecting a mask based on a threshold is computationally efficient, this method is relatively naive compared to other SFT methods that dynamically choose parameters during fine-tuning, which may lead to lower performance. Although it was compared with LT-SFT, there is a lack of comparison with other SFT methods.
- In the table, it is unclear if this method offers a significant performance improvement. While the FID scores are mostly favorable, the CLIP scores appear to be higher for LoRA. Additionally, the L_rank removal in the ablation study does not lead to a significant performance change.

**Questions:**

- Would similar results be achievable if other SFT methods stored the selected parameters in leaf nodes and updated them as implemented here?
- Could you provide more explanation regarding the impact of training process randomness on initially inefficient parameters becoming efficient?

---

> ### Author Response · Authors · 2024-11-23
> **Response to Reviewer bh95 (1/3)**
>
> We sincerely thank you for the valuable suggestions to improve our work. The following content provides a point-by-point response to address your concerns. The corresponding modifications have been refined in the revised submission, highlighted in blue.
>
> **Q1: Although the paper differentiates itself from selective fine-tuning, I think this method still appears to be a form of selective fine-tuning. The memory efficiency improvement seems to stem from implementing a separate node for gradient storage and selectively fine-tuning only those nodes, rather than from an inherent algorithmic difference.**
>
> **A1:** Thanks for your question. Our method can be interpreted as a selective fine-tuning method and this is not denied in the paper. Nevertheless, the differences between ours and conventional selective fine-tuning methods include: **1)** we select a different subset of parameters ("unimportant" parameters) to finetune, and **2)** our unstructual backpropagation strategy, which is proposed to enhance the GPU memory efficiency by designing separate nodes. This strategy is not only limited to our method, but can be extended to other selective PEFT methods, as illustrated in Lines 408~410 of the main paper. **3)** To make it clearer, we add more relative descriptions in the end of Sec. 4.4:” Notably, our unstructual backpropagation strategy is not limited to our method only. It can be employed in other SFT methods like LT-SFT, which can advance the development of future SFT fields.”
>
> **Q2: Selecting parameters based on a specific threshold seems not a new concept. For example, the related work PaFi is described that it also trains based on absolute values.**
>
> **A2:** Thanks for your question. There are several differences between PaFi and Our SaRA. **1) Motivation:** The motivation between our method and PaFi is quite different, PaFi only aims to fix the parameters before training, therefore, it selects the trainable parameters based on the parameter values. But it does not analyze why these parameters can be used. In contrast, our SaRA first conducts extensive experiments to validate the temporal inefficiency and potential effectiveness of the selected parameters. And then this observation motivates us to make use of the parameters with the smallest absolute values. **2) Technical contribution:** Beyond simply using these parameters, we further proposed a low-rank constraint and progressive training scheme, along with the unstructual backpropagation strategy, which effectively improves the model performance and training efficiency.
>
> **Q3: The reasoning behind inefficient parameters becoming efficient during training due to the randomness of the training process is unclear. Since the initial weights are set randomly, many values are likely to change through training. There is no strong basis to assume a correlation with the initial values. Even parameters with initially small values are expected to converge toward the average distribution, making Figure 3 somewhat self-evident.**
>
> **A3:** Thanks for your question. **1)** In Fig. 3, the initial weights come from a pre-trained diffusion model on FFHQ, rather than a random distribution. We continue training it and record the changes of all the parameters in Fig.3. It shows that most of the ineffective parameters in the pre-trained model become effective again, validating their potential effectiveness. **2)** This positive transformation from ineffective to effective is ensured by the backpropagation of training loss gradients. Specifically, the fine-tuned parameters are adapted and trained on a specific training dataset for a particular task.

---

> > ### Author Response · Authors · 2024-11-23
> > **Response to Reviewer bh95 (2/3)**
> >
> > **Q4: Although selecting a mask based on a threshold is computationally efficient, this method is relatively naive compared to other SFT methods that dynamically choose parameters during fine-tuning, which may lead to lower performance. Although it was compared with LT-SFT, there is a lack of comparison with other SFT methods.**
> >
> > **A4:** Thanks for your question. **1)** In image generation tasks, the two goals for fine-tuning are for the model to learn target-domain knowledge, while maintaining its prior knowledge. For the previous selective PEFT methods, like LT-SFT and DiffPruning [1], they aim to choose the most effective parameters for the downstream tasks, which may potentially cause the model to lose the prior knowledge, therefore resulting in a lower CLIP Score. **2)** To validate this, we further compare the average metric on five downstream datasets among our model (5M parameters) , LT-SFT (5M parameters), and DiffPruning as below (detailed results are supplemented in Sec. C Tab. 4). It can be seen that although we pre-define the parameter mask, our model still gets the best FID score, indicating that our model learns the most target-domain knowledge. And since we only fine-tune the initial ineffective parameters, the prior knowledge is well-kept, which results in a better CLIP score than LT-SFT. For Diffprune, it is obvious that the model does not learn much target-domain knowledge, therefore resulting in an extremely high FID.
> >
> > | SD 1.5     | DiffPruning | LT-SFT | SaRA (Ours) |
> > | :---------- | :-----------: | :------: | :-----------: |
> > | FID        | 206.14      | 165.72 | **147.80**      |
> > | CLIP Score | **28.86**       | 26.37  | 27.73       |
> > | VLHI       | 1.00        | 0.27   | **1.44**        |
> >
> > **Q5: In the table, it is unclear if this method offers a significant performance improvement. While the FID scores are mostly favorable, the CLIP scores appear to be higher for LoRA. Additionally, the L_rank removal in the ablation study does not lead to a significant performance change.**
> >
> > **A5:** Thanks for your question. **1)** For image generation tasks, FID and CLIP scores exhibit a certain degree of mutual exclusivity in finetuning a text-to-image model to downstream tasks (*i.e.*, a model that does not learn any downstream knowledge will result in the best CLIP but the worst FID score), and this is why we introduce a new metric VLHI to balance both the two metrics (as illustrated in Lines 725-727). As shown in Tab. 1, our model achieves the best VLHI, indicating the best performance in learning the target-domain knowledge while keeping the model priors. **2)** Moreover, the model without $L_{rank}$ results in the worst CLIP score, indicating a significant performance drop. We further conduct additional ablation studies on Cyberpunk and ElementFire datasets, the results are shown below, where we can see that $L_{rank}$ can prevent the model from decreasing the CLIP score too much (results are supplemented in Sec. H-Tab. 9).
> >
> > | Dataset            | Cyberpunk | Cyberpunk | ElementFire | ElementFire |
> > | :------------------ | :---------: | :-----------: | :-----------: | :------: |
> > | Metric             | FID       | CLIP        | FID         | CLIP   |
> > | SaRA               | 121.67    | 27.30       | 132.67      | 26.77 |
> > | SaRA w/o. $L_{rank}$ | 120.33    | 26.52       | 131.56      | 25.88  |
> >
> >
> >
> > **Q6: Would similar results be achievable if other SFT methods stored the selected parameters in leaf nodes and updated them as implemented here?**
> >
> > **A6:** Thanks for your question. **1)** Unstructural backpropagation (UB) strategy is proposed to enhance the memory efficiency for SFT methods, which is a plug-and-play general method. It can achieve a similar effect for other SFT methods that decrease the GPU memory more significantly while obtaining similar results. **2)** We employ our unstructural backpropagation strategy on LT-SFT (50M parameters, SD2.0) and show the comparison results in the following table. It can be seen that the model performance is not influenced, while largely decrease the memory cost.
> >
> > | Dataset             | Barbie | Cyberpunk | ElementFire | Expedition | Hornify | Mean   |
> > | :------------------- | :------: | :---------: | :-----------: | :----------: | :-------: | :------: |
> > | $\Delta$ FID        | +0.21% | -0.13%    | -0.45%      | +0.07%     | +0.28%  | -0.02% |
> > | $\Delta$ CLIP Score | +0.23% | -0.08%    | -0.39%      | +0.04%     | +0.32%  | +0.12% |
> > | $\Delta$ Memory     | -9.2GB | -9.2GB    | -9.2GB      | -9.2GB     | -9.2GB  | -9.2GB |

---

> > > ### Author Response · Authors · 2024-11-23
> > > **Response to Reviewer bh95 (3/3)**
> > >
> > > **Q7: Could you provide more explanation regarding the impact of training process randomness on initially inefficient parameters becoming efficient?**
> > >
> > > **A7:** Thanks for your question. For a continuous model, due to random data sampling and the stochastic nature of the noise addition and denoising processes in diffusion models, it is reasonable to assume that the probability of any given parameter's gradient update increasing or decreasing is uniform. Assume that each parameter's gradient update $\Delta$ $\theta$  is an independent random variable, and the probability of increasing or decreasing is uniform (i.e., symmetric distribution). We can then assume that the expected value \( $\mu$ \) of these gradient updates is 0, and the variance is \( $\sigma^2$ \).
> > >
> > > For N gradient updates, the cumulative effect is:
> > > $$
> > > \theta = \sum_{i=1}^N \Delta \theta_i
> > > $$
> > > According to the Central Limit Theorem, when \( N \) is large, the cumulative gradient update \($\theta$) will approach a normal distribution:
> > > $$
> > > \theta \sim \mathcal{N}(0, N\sigma^2)
> > > $$
> > > Therefore, each parameter is randomly located around 0, and as the training process goes on, it will become effective again (jump out of the threshold).
> > >
> > > **Reference**
> > >
> > > [1] Guo D, Rush A M, Kim Y. Parameter-efficient transfer learning with diff pruning[J]. arXiv preprint arXiv, 2020.

---

> > > > ### Comment · Reviewer_bh95 · 2024-11-26
> > > >
> > > > Thank you for providing detailed responses to my initial review and questions. Most of my concerns have been addressed, so I will raise my score. I was particularly curious about comparisons with other SFT methods, and the detailed explanations and experiments helped highlight the unique aspects of SARA. However, while it is more memory-efficient than LoRA, it requires several additional training techniques and shows slight performance degradation. Additionally, I felt it was unclear whether its quality was significantly better than other SFT methods based on some visual results. Therefore, I raised my score to borderline accept.

---

> > > > > ### Author Response · Authors · 2024-11-28
> > > > >
> > > > > Dear Reviewer bh95,
> > > > >
> > > > > Thank you for your positive and encouraging feedback on our work. We are delegated to hear that our responses have addressed most of your concerns. We are grateful for the time and effort you have invested.
> > > > >
> > > > > For your concerns on the visual results between our method and other STF methods, we have further compared our method with more SFT methods, including LT-SFT, FishMask, Diffpruning, and an ablated method that fine-tunes the largest parameters on Stable Diffusion 1.5 with 50M trainable parameters. The qualitative comparison results are shown in Fig.18.
> > > > >
> > > > > It can be observed that LT-SFT does not learn the target style well in the ElementFire and Horinfy datasets. FishMask tends to generate artifacts as it tunes some effective parameters in the pretrained weights, disrupting part of the model priors. DiffPruning fails to capture task-specific information, resulting in outputs that differ significantly from the target style (despite the fact that we have tried different hyperparameters). Additionally, the ablated model that fine-tunes the largest parameters tends to overfit, similar to the full-parameter fine-tuning model. Since the most important parameters are all fine-tuned, it is prone to overfitting to the target domain, leading to generated images that do not align well with the given prompts. In contrast, our SaRA fits the five datasets well while preserving the model priors, indicating superior performance among the different selective PEFT methods. (For more details, please refer to Sec. C and Fig.18 in the revised version)
> > > > >
> > > > > Best regards,
> > > > >
> > > > > The Authors

---

### Official Review · Reviewer_GkU7 · 2024-11-03

**Soundness:** 4
**Presentation:** 4
**Contribution:** 4
**Rating:** 8
**Confidence:** 5

**Summary:**

The paper presents a new method for fine-tuning diffusion models by training only low-value parameters making them effective on a new task. Additionally, a nuclear norm is used to prevent overfitting, and efficient selective backpropagation and progressive parameter adjustment reduce the memory and time requirements during training. The results show SaRA is on par or better than other fine-tuning methods in terms of FID, CLIP score, and qualitative assessment.

**Strengths:**

1. The paper is very well written and easy to follow. It has a good flow of information. Every claim is carefully explained and proven by experiments or analysis.
2. The novelty of the method is good. The idea of fine-tuning only ineffective parameters was explored before but combined with nuclear norm regularization, novel approach to backpropagation of a sparse matrices, and adaptive fine-tuning, creates a valuable addition to the field.
3. The experiments are extensive, both comparisons and ablation study.
4. Qualitative results suggest an improvement over other methods.
5. Quantitative results show the model behaves better or similar to other methods. I can see it becoming one of the methods of choice depending on one's needs.

**Weaknesses:**

1. Minor - as mentioned in Strengths 5., the results are not showing overall superiority over other methods.

**Questions:**

1. The authors should consider splitting Figures 2 b) and 5 into more subplots. In my opinion, they are too cluttered and it takes too much time to read them.
2. In Table 1, the use of "optimal" and "sub-optimal" is not correct, e.g. optimal FID is 0. "Best" and "second best" or something similar should be used.
3. Could the authors provide more qualitative results for the same prompts (Appendix C) with different seeds to see the diversity of the samples?
4. It would be good to include some of the visual results in the main text.
5. Can authors elaborate on how CLIP score is related to overfitting?

---

> ### Author Response · Authors · 2024-11-23
> **Response to Reviewer GkU7**
>
> We sincerely thank you for the valuable suggestions to improve our work. The following content provides a point-by-point response to address your concerns. The corresponding modifications have been refined in the revised submission, highlighted in blue.
>
> **Q1: Results are not showing overall superiority over other methods.**
>
> **A1:** Thanks for your question. For image generation tasks, FID and CLIP scores exhibit a certain degree of mutual exclusivity in finetuning a text-to-image model to downstream tasks (i.e., an overfitted model will result in the best FID but the worst CLIP score), and this is why we introduce a new metric VLHI to balance both the two metrics (as illustrated in Lines 725-727). Evaluated by VLHI, our model achieves the best performance in learning the task-specified knowledge and keeping the model prior.
>
> **Q2: The authors should consider splitting Figures 2 b) and 5 into more subplots. In my opinion, they are too cluttered and it takes too much time to read them.**
>
> **A2:** Thanks for you suggestion. We have splitted Figure 2 (b) and Figure 5 into subplots, which is more clear to read.
>
> **Q3: In Table 1, the use of "optimal" and "sub-optimal" is not correct, e.g. optimal FID is 0. "Best" and "second best" or something similar should be used.**
>
> **A3:** Thanks for your reminding. We have corrected them in Tab. 1, along with some corresponding corrections in the main text.
>
> **Q4: Could the authors provide more qualitative results for the same prompts (Appendix C) with different seeds to see the diversity of the samples?**
>
> **A4:** Thanks for your suggestion. We have included more visualization results in Sec. C (Figs. 11-13), which show the diverse generation capabilities of our model.
>
> **Q5: It would be good to include some of the visual results in the main text.**
>
> **A5:** We have included Fig. 7 (Part of Fig. 8 in Appendix) in the main paper. Since the main manuscript contains quite many quantitative experiments, we can only show part of the visual results limited by number of pages. Thanks for your understanding, and we have also illustrated this Lines 499-501 in Sec. 5.2.
>
> **Q6: Can authors elaborate on how CLIP score is related to overfitting?**
>
> **A6:** Thanks for your question. If the model is overfitted, the model can only generate the images that resemble the images in training dataset, whose output images do not align well with the given texts. In this case, the CLIP score will be extremely low. So, a highly overfitted model will result in a very low CLIP score. We also add a detailed explanation in Sec. B.

---

> > ### Comment · Reviewer_GkU7 · 2024-11-25
> >
> > I thank the authors for addressing my questions. After reading other reviews and authors' responses, I decided to keep my score, as there are no major unresolved concerns.

---

> > > ### Author Response · Authors · 2024-11-27
> > >
> > > Dear Reviewer GkU7,
> > >
> > > Thank you for your kind and encouraging feedback on our work! We’re delighted to know that our rebuttal has addressed your concerns. We deeply value the time and effort you’ve dedicated to providing detailed reviews and insightful suggestions, which have greatly contributed to enhancing our work.
> > >
> > > Best regards,
> > >
> > > The Authors

---

### Official Review · Reviewer_EfNM · 2024-11-04

**Soundness:** 2
**Presentation:** 3
**Contribution:** 3
**Rating:** 5
**Confidence:** 4

**Summary:**

To leverage the unimportant parameters concept in model pruning, this paper proposes a model fine-tuning method for diffusion model by reusing these ineffective parameters. The authors find that such ineffective parameters with small absolute values are random and dynamically change over finetuning. Based on this observation, they further design some efficient strategies for tuning these parameters.

**Strengths:**

1. This paper verifies that the ineffective parameter concept also applies for diffusion model, i.e., the parameters with small absolute values are not important for the generation.
2. This paper adopts a series of strategies for the efficient tuning of these ineffective parameters.
3. The effectiveness of this method is verified on the stable diffusion series. It demonstrates better performance over the baselines.

**Weaknesses:**

1. The main limitation is that this paper poorly extends the concept in static model to the finetuning area, which is a dynamic model. Specifically, the ineffective concept works in model pruning, which is a given fixed model. The ineffective parameters in such static model can be discarded or reused. However, in this paper, the model parameters dynamically change during finetuning, and unimportant parameters also change. Thus, a right way to extend such unimportant parameters is to study their dynamics over change, instead of simple reuse.  Simple reuse may have several issues, for example, smaller optimal parameter search space in the finetuning case.
2. Another concern may lie in how to merge multi tasks’ parameters, and their merging performance. The multi-task parameter merging is a good property of LORA. It is encouraged to explain and verify this.

**Questions:**

How to understand the better performance after setting some parameters to 0 in Section 3.1? This is actually interesting and may be useful for understanding the behavior of diffusion model.

---

> ### Author Response · Authors · 2024-11-23
> **Response to Reviewer EfNM**
>
> We sincerely thank you for the valuable suggestions to improve our work. The following content provides a point-by-point response to address your concerns. The corresponding modifications have been refined in the revised submission, highlighted in blue.
>
> **Q1: This paper poorly extends the concept in static model to the finetuning area.**
>
> **A1:** Thanks for your suggestion. **1)** We have analyzed the dynamic change of these unimportant parameters in Fig. 3 (Sec. 3.2), where we find that these ineffective parameters are not born to be ineffective, but come from the unstable training process. After finetuning the model, these parameters can be effective again. **2)** Moreover, in Sec. 4.3, we dynamically adjust the parameters to be trained, which is designed to further improve the effectiveness of our learned parameters. **3)** Meanwhile, we find that these sparse parameters even have a larger feature space than LoRA as analyzed in Sec. G, where SaRA can learn more task-specified knowledge, (which is also validated by the FID score in Tab. 1), indicating a larger parameter search space in our model.
>
> **Q2: How to merge multi tasks’ parameters, and their merging performance.**
>
> **A2:** Thanks for your reminding. The multi-task parameter merging is indeed an important property of LoRA. Comparably, we propose a new parameter combination method to combine different parameters learned from different datasets in Sec. G. Specifically, for two SaRA parameters $\Delta P_1$ and $\Delta P_2$ learned from different datasets, we can merge them by:
> $$
> \Delta P = \alpha_1\Delta P_1 + \alpha_2\Delta P_2
> $$
>
> In this way, the merged SaRA parameter $\Delta P$ will contain the knowledges from both the datasets, whose output image contains both the target features. The visualization results can be seen in Fig. 16 and we also include this parameter merging technique in Sec. G.
>
> **Q3: How to understand the better performance after setting some parameters to 0 in Section 3.1? This is actually interesting and may be useful for understanding the behavior of diffusion model.**
>
> **A3:** Thanks for your question. 1) In Sec 3.2, we conduct exploration experiments and empirically find that the parameters close to 0 are caused by the unstable training process. We argue that these parameters may introduce some harmful randomness into the generated results. After setting these parameters to 0, the model would not be influenced by them, therefore it may improve the FID score a little bit. 2) Also, some pruning studies [1,2] also find that removing unnecessary weights can reduce the model's complexity and decrease the risk of overfitting. Setting these parameters to zero can suppress noise, allowing the model to focus more on important features, thereby improving its performance.
>
>
> **Reference**
>
> [1] Han S, Pool J, Tran J, et al. Learning both weights and connections for efficient neural network[J]. NeurIPS, 2015.
>
> [2] Choudhary T, Mishra V, Goswami A, et al. A comprehensive survey on model compression and acceleration[J]. Artificial Intelligence Review, 2020.

---

> ### Author Response · Authors · 2024-11-27
>
> Dear Reviewer EfNM,
>
> We greatly appreciate the time and effort you have devoted to reviewing our paper!
>
> Since there is only one day left to upload the revised PDF, we kindly hope you might find a moment to review our response, if your schedule allows. Should there be any further points requiring clarification or improvement, we remain fully committed to addressing them without delay. Thank you once again for your valuable feedback on our work!
>
> Best regards,
>
> The Authors

---

> ### Author Response · Authors · 2024-12-01
>
> Dear Reviewer EfNM,
>
> We sincerely apologize for our repeated requests and any inconvenience they may have caused. We truly appreciate your time and effort in reviewing our paper, and we understand that you have many other commitments.
>
> As the discussion period is coming to a close, we kindly ask if you might be able to provide your feedback at your earliest convenience. If there are any further points that require clarification or revision, we are fully committed to addressing them promptly.
>
> Thank you once again for your patience and invaluable feedback.
>
> Best regards,
>
> The Authors

---

### Official Review · Reviewer_RosU · 2024-11-04

**Soundness:** 3
**Presentation:** 3
**Contribution:** 3
**Rating:** 6
**Confidence:** 3

**Summary:**

The paper presents a novel approach for fine-tuning pre-trained diffusion models called SaRA for visual content generation. The method builds on a key insight: parameters with the smallest absolute values in diffusion models contribute minimally to generation due to training instabilities, allowing for their selective reuse. SaRA enhances these low-impact parameters by applying a sparse weight matrix that learns task-specific knowledge while retaining the model’s generalization abilities. To avoid overfitting, the authors introduce a nuclear-norm-based low-rank training scheme. Additionally, SaRA includes a progressive parameter adjustment strategy and an unstructured backpropagation approach to efficiently manage memory use during fine-tuning.


Note: the supplementary materials contain the author's username and IP address (Supplementary Material/code/.idea/deployment.xml)

**Strengths:**

1. The method visualizations (Figures 1 and 4) provide a step-by-step comparison of the proposed approach with previous techniques that helps the reader to easily understand the nuances of SaRA.
2. The authors show that the method could be applied to different diffusion models denoisers architectures U-Net (SD1.5 and 2.0) and Diffusion Transformer (SD3.0).
3. The design of the method allows a plug-and-play experience for the users that is highly beneficial for practical adoption.
4. The authors demonstrate the capabilities of the proposed approach on various widely-used by the open source community datasets.

**Weaknesses:**

1. The authors introduce a novel Visual-Linguistic Harmony Index (VLHI) metric; however, it's described only in the appendix.
2. No comparison with the recent SOTA PEFT techniques (e.g., DoRA [Liu et al. 2024] that is available for different models on mentioned in the paper CIVITAI).
3. No ablation on scaling the trained SaRA weights for the inference (as the lora_scale parameter controls the influence of LoRA weights) or mentioning it in the limitations.
4. The authors say that for the FID computation they sampled 5K images from the source and generated data; however, BarbieCore dataset has only 315 images which is definitely not enough for the proper FID evaluation. The details about the sizes of the used datasets should be in the paper.
5.  CLIP L/14 used by authors is trained to provide overall image captions and could miss the details. The visual language models-based evaluations used in T2I-CompBench++[Huang et al. 2023] could be more accurate.
6. The authors skip the most popular Stable Diffusion XL 1.0 version, whereas they include 2.0.
7. Typographical mistakes such as:
*) Table 1: wrong column 2&3 names
*) Figure 1: addictive-> additive

**Questions:**

see the weaknesses section

---

> ### Author Response · Authors · 2024-11-23
> **Response to Reviewer RosU (1/2)**
>
> We sincerely thank you for the valuable suggestions to improve our work. The following content provides a point-by-point response to address your concerns. The corresponding modifications have been refined in the revised submission, highlighted in blue.
>
> **Q1: The authors introduce a novel Visual-Linguistic Harmony Index (VLHI) metric; however, it's described only in the** **Appendix.**
>
> **A1:** Thanks for your suggestion. Considering the constraints of the paper's length, we describe the novel VLHI metric in Sec.B in Appendix. Following your advice, we have moved its definition from the supplementary material to the beginning of the experiment section (Sec. 5) for better presentation.
>
> **Q2: No comparison with the recent SOTA PEFT techniques (e.g., DoRA [Liu et al. 2024] that is available for different models on mentioned in the paper CIVITAI).**
>
> **A2:** Thank you for your suggestion. We conducted further comparisons of our model with DoRA [1] using SD 1.5 and SD 2.0. Detailed results can be found in Sec. C of the Appendix. The key metrics are summarized in the following table for DoRA and SaRA with different parameter numbers (e.g., SaRA-50M refers to SaRA with 50M parameters). The results show that DoRA performs similarly to LoRA but fails to achieve as good an FID as our model, indicating that it captures less target-domain knowledge compared to our approach. Although DoRA achieves slightly better CLIP scores under certain conditions, the superior VLHI achieved by our model highlights its ability to effectively learn target-domain knowledge while maintaining the original model's prior.
>
> | SD 1.5 | DoRA-50M | SaRA-50M (Ours) | DoRA-20M | SaRA-20M (Ours) | DoRA-5M | SaRA-5M (Ours) |
> | :---------- | :--------: | :---------------: | :--------: | :---------------: | :-------: | :--------------: |
> | FID        | 146.22   | **134.96**          | 145.55   | **136.92**          | 149.07  | **147.80**         |
> | CLIP Score | **27.82**    | 27.08           | **27.73**    | 27.53          | 27.70   | **27.73**          |
> | VLHI       | 1.53     | **1.55**            | 1.51     | **1.69**            | 1.38    | **1.44**           |
>
> | SD XL      | DoRA-50M | SaRA-50M (Ours) | DoRA-20M | SaRA-20M (Ours) | DoRA-5M | SaRA-5M (Ours) |
> | :---------- | :--------: | :---------------: | :--------: | :---------------: | :-------: | :--------------: |
> | FID        | 154.20   | **137.45**          | 153.68   | **144.46**        | 153.98  | **147.26**         |
> | CLIP Score | **28.82**    | 28.45           | 28.71    | **28.84**           | 28.59   | **28.94**          |
> | VLHI       | 1.06     | **1.71**           | 1.03     | **1.58**            | 0.94    | **1.44**           |
>
> **Q3: No ablation on scaling the trained SaRA weights for the inference (as the lora_scale parameter controls the influence of LoRA weights) or mentioning it in the limitations.**
>
> **A3:** Thanks for your suggestion. The scaling weight is indeed an important property for the PEFT methods in generative models. We conduct additional experiments on different SaRA weights and the visualization results can be seen in **Sec. F of Appendix**. As the scaling weight grows, SaRA further emphasizes the target-domain knowledge, while sacrificing the generalization ability. When the weight is too big, the model tends to generate some artifacts. This phenomenon is quite similar to LoRA-Scale.
>
> **Q4: The authors say that for the FID computation they sampled 5K images from the source and generated data; however, BarbieCore dataset has only 315 images which is definitely not enough for the proper FID evaluation. The details about the sizes of the used datasets should be in the paper.**
>
> **A4:** Thanks for your suggestion. Although the FID may not be so accurate when the image number is limited, but it can still be used as a relative comparison, which is widely used in few-shot (less than 1000 images) domain adaptation tasks [2,3,4,5]. Compared to the previous method, our approach achieves the best FID score at most of the conditions, demonstrating the effectiveness of SaRA. Moreover, We have provided additional detailed explanations of the datasets in Sec. B of the Appendix, where the size of each dataset can also be found in the following table. Although there are only hundreds of images, the dataset size is still larger than the few-shot datasets [2,3,4,5].
>
> | Dataset      | Barbie | Cyberpunk | ElementFile | Expedition | Hornify |
> | :------------ | :------: | :---------: | :-----------: | :----------: | :-------: |
> | Image Number | 316    | 440       | 156         | 396        | 236     |

---

> ### Author Response · Authors · 2024-11-23
> **Response to Reviewer RosU (2/2)**
>
> **Q5: CLIP L/14 used by authors is trained to provide overall image captions and could miss the details. The visual language models-based evaluations used in T2I-CompBench++[Huang et al. 2023] could be more accurate.**
>
> **A5:** Thanks for your suggestion. We have evaluated the model performance by the visual language model in T2I-CompBench++ for the models trained on stable diffusion 1.5 (the metric is denoted as B-VQA), and the detailed results are shown in Tab. 6 of the appendix. We summarize the results in the following table, where for each method, we compute the average FID, B-VQA and VLHI across all the datasets and model scales (5M, 20M, and 50M parameters). It shows that our model achieves the best performance across all three metrics.
>
> | SD 1.5 | DoRA   | LoRA   | Adaptformer | LT-SFT | SaRA (Ours) |
> | :------ | :------: | :------: | :-----------: | :------: | :-----------: |
> | FID    | 146.92 | 150.36 | 151.25      | 157.13 | **139.96**  |
> | B-VQA  | 0.46   | 0.48   | 0.48        | 0.48   | **0.48**        |
> | VLHI   | 1.19   | 1.24   | 1.22        | 1.08   | **1.61**        |
>
> **Q6: The authors skip the most popular Stable Diffusion XL 1.0 version, whereas they include 2.0.**
>
> **A6:** We have conducted experiments on Stable Diffusion XL 1.0, comparing DoRA, Lora, LT-SFT, Adaptformer, and our model. The detailed results are provided in Tab.5 and Sec. C in the Appendix. Similar to the table above, we compute the average FID, CLIP score, and VLHI across all the datasets and model scales, and show the results in the following table. It can be seen that our model achieves the best performance across these 3 metrics, validating the effectiveness and robustness of our model.
>
> | SD XL      | DoRA   | LoRA   | Adaptformer | LT-SFT | SaRA (Ours) |
> | :---------- | :------: | :------: | :-----------: | :------: | :-----------: |
> | FID        | 153.96 | 145.84 | 151.72      | 150.16 | **143.39**      |
> | CLIP Score | 28.70  | 28.57  | 28.45       | 28.50  | **28.74**       |
> | VLHI       | 1.01   | 1.35   | 0.97        | 1.09   | **1.58**        |
>
> **Q7: Typographical mistakes such as: \*) Table 1:**wrong column 2&3 names** ) Figure 1: addictive-> additive**
>
> **A7:** Thanks for your suggestion. We have carefully revised the re-submission.
>
>
> **Reference**
>
> [1] Liu S, Wang C Y, Yin H, et al. DoRA: Weight-Decomposed Low-Rank Adaptation[C]. ICML, 2024
>
> [2] Gal R, Patashnik O, Maron H, et al. Stylegan-nada: Clip-guided domain adaptation of image generators[J]. ACM Transactions on Graphics (TOG), 2022.
>
> [3] Ojha U, Li Y, Lu J, et al. Few-shot image generation via cross-domain correspondence[C]//CVPR, 2021.
>
> [4] Mo S, Cho M, Shin J. Freeze the discriminator: a simple baseline for fine-tuning gans[J]. arXiv, 2020.
>
> [5] Wang Y, Gonzalez-Garcia A, Berga D, et al. Minegan: effective knowledge transfer from gans to target domains with few images[C]//CVPR, 2020.

---

> ### Author Response · Authors · 2024-11-27
>
> Dear Reviewer RosU,
>
> We greatly appreciate the time and effort you have devoted to reviewing our paper!
>
> Since there is only one day left to upload the revised PDF, we kindly hope you might find a moment to review our response, if your schedule allows. Should there be any further points requiring clarification or improvement, we remain fully committed to addressing them without delay. Thank you once again for your valuable feedback on our work!
>
> Best regards,
>
> The Authors

---

> ### Author Response · Authors · 2024-12-01
>
> Dear Reviewer RosU,
>
> We sincerely apologize for our repeated requests and any inconvenience they may have caused. We truly appreciate your time and effort in reviewing our paper, and we understand that you have many other commitments.
>
> As the discussion period is coming to a close, we kindly ask if you might be able to provide your feedback at your earliest convenience. If there are any further points that require clarification or revision, we are fully committed to addressing them promptly.
>
> Thank you once again for your patience and invaluable feedback.
>
> Best regards,
>
> The Authors

---

### Public Comment · ~Kartikeya_Bhardwaj1 · 2024-11-13
**Related Work: Sparse High Rank Adapters (Neurips 2024)**

Dear Authors,

Thanks for the interesting work SARA. It is important for the research community to focus on sparse finetuning techniques that are as memory-efficient as LoRA so we really appreciate your contributions. We recently published a highly related work at Neurips 2024 "Sparse High Rank Adapters" (SHiRA) which also presents memory benefits using a PEFT implementation compared to LoRA and analyzes multi-adapter fusion properties for sparse finetuning (see https://openreview.net/forum?id=6hY60tkiEK, older preprint: https://arxiv.org/abs/2406.13175). Can you please discuss our _concurrent_ work in your related work section?

Another important related work is SpIEL. https://arxiv.org/abs/2401.16405.

It would be nice to see qualitative differences between these recent works (SARA, SHiRA, and SpIEL).

All the best!

---

> ### Author Response · Authors · 2024-11-24
> **Response to Kartikeya Bhardwaj**
>
> Thank you for the reminder.
>
> We have included and discussed SHiRA and SpIEL in the related work section. Briefly, the comparison between SHiRA and our SaRA is as follows: **1)** SHiRA focuses on improving adaptation ability through a high-rank sparse matrix, while our SaRA aims to preserve model priors while effectively learning target-domain information. Therefore, these two works differ in their objectives and contribute distinct technical advancements. **2)** Regarding the memory improvement strategy, we modify the optimizer such that only a single line of code is required to call AdamW-SaRA, which efficiently updates specific parameters and gradients (Sec. B in Appendix). In contrast, SHiRA uses a hook mechanism to achieve a similar purpose. However, due to the lack of open-source code, we cannot perform further quantitative or qualitative comparisons. We would appreciate more details about your code implementation, as a combination of the strengths of both approaches may lead to a more efficient and enhanced version of the training process.
>
> Thanks for your attention to our work!

---

### Public Comment · ~Shitong_Shao1 · 2024-11-22
**LISA and SaRA [1/3]**

Thank you for your interesting work! I believe the two key issues encountered when fine-tuning a diffusion model are:

1.  The excessive memory overhead required by the optimizer to store the first and second-order moments in both AdamW and AdamW8bit.

2.  Additional memory demands arising from models like DPO and Consistency Model, which require the introduction of an additional EMA update. In other words, the algorithm proposed by the authors is not an algorithm specifically designed around the diffusion model.

Additionally, this algorithm is quite similar to LISA [1]. I personally experimented with LISA on a diffusion model, and the results were quite good, though it imposed a significant memory burden. Interestingly, I found that removing the first and second-order moments for parameters that do not need updating did not affect performance and significantly reduced the memory footprint.

[1] LISA: Layerwise Importance Sampling for Memory-Efficient Large Language Model Fine-Tuning, NeurIPS 2024.

I present the implementation of LISA in diffusion model here, everyone can try this:

```python
import torch
import numpy as np
import gc

class LISADiffusion:
    def __init__(self, model, other_model=None, rate=None, dtype=torch.float16,
                 grad_aware=False, accelerator=None):
        self.model = model
        self.other_model = other_model
        self.dtype = dtype
        self.grad_aware = grad_aware
        if self.other_model is not None:
            for param in self.other_model.parameters():
                param.requires_grad = True
        self.rate = rate
        self.accelerator = accelerator
        self.last_epoch = 0
        num_sort = []
        name_sort = []
        total_num = 0.
        lisa_target_name = self.get_better_parameters_name
        for i, (name, param) in enumerate(list(model.named_parameters())[1:-1]):
            num_sort.append([i+1, param.numel()])
            total_num += param.numel()
            name_sort.append(name)
        self.allowed_index = [j[0] for j in sorted(num_sort,key=lambda x:x[1],reverse=False)]
        self.probability = [1. for j in range(len(self.allowed_index))]
        # used_num = 0.
        for i, j in enumerate(self.allowed_index):
            if any([(module_key in name_sort[j-1]) for module_key in lisa_target_name]):
                self.probability[j-1] = (0.9995 ** i)
            else:
                self.probability[j-1] = 0
        p_sum = sum(self.probability)
        self.probability = [j/p_sum for j in self.probability]
        self.initialize()

    def freeze_all_layers(self, model):
        for param in model.parameters():
            param.requires_grad = False

    def random_activate_layers(self, model, p):
        activate_number = int((len(list(model.parameters()))-2) * p)
        index = np.random.choice(range(1,len(list(model.parameters()))-1,1), activate_number, replace=False, p=self.probability)
        count = 0
        for param in model.parameters():
            if count == 0 or count == len(list(model.parameters()))-1:

                param.requires_grad = True
                param.data = param.data.to(dtype=torch.float32)
            elif count in index:
                param.requires_grad = True
                param.data = param.data.to(dtype=torch.float32)
            else:
                param.requires_grad = False
                param.data = param.data.to(dtype=self.dtype)
            count += 1
        count_number = 0
        pos_count_number = 0
        for param in model.parameters():
            if param.requires_grad:
                pos_count_number += param.numel()
            count_number += param.numel()
        if torch.distributed.get_rank() == 0:
            print("Total Trainable Parameters:", round(pos_count_number*100/count_number,2),"%")

    def lisa(self, model, p=0.25):
        self.freeze_all_layers(model)
        self.random_activate_layers(model, p)
```

---

> ### Public Comment · ~Shitong_Shao1 · 2024-11-22
> **LISA and SaRA [2/3]**
>
> ```python
>     def lisa_recall(self):
>         param_number = len(list(self.model.parameters()))
>         lisa_p = 8 / param_number if self.rate is None else self.rate
>         self.lisa(model=self.model,p=lisa_p)
>         self.last_epoch += 20
>         self.model.zero_grad()
>         if self.accelerator is not None:
>             self.accelerator.clear()
>         gc.collect()
>         torch.cuda.empty_cache()
>
>         if self.grad_aware:
>             new_probability = []
>             for i, param in enumerate(list(self.model.parameters())[1:-1]):
>                 if self.grad_record.get(param, None) is None or len(self.grad_record[param]) < 200:
>                     v = 1. if len(new_probability) == 0 else max(new_probability)
>                 else:
>                     v = torch.stack(self.grad_record[param],0).std().item() + 1e-6
>                 new_probability.append(v)
>             new_probability = [j/sum(new_probability) for j in new_probability]
>             for i, new_p in enumerate(new_probability):
>                 self.probability[i] = self.probability[i] * 0.01 + new_p * 0.99
>             self.probability = [j/sum(self.probability) for j in self.probability]
>
>
>     def initialize(self):
>         self.optimizer_dict = dict()
>         self.scheduler_dict = dict()
>         self.grad_record = dict()
>
>     def register(self, optimizer_class=None, get_scheduler=None, accelerator=None,
>                  optim_kwargs={}, sched_kwargs={}):
>
>         self.lisa_recall()
>         for p in self.model.parameters():
>             self.grad_record[id(p)] = []
>             if p.requires_grad:
>                 self.optimizer_dict[id(p)] = optimizer_class([{"params":p}], **optim_kwargs)
>                 if accelerator is not None:
>                     self.optimizer_dict[id(p)] = accelerator.prepare_optimizer(self.optimizer_dict[id(p)])
>
>         for p in self.model.parameters():
>             if p.requires_grad:
>                 self.scheduler_dict[id(p)] = get_scheduler(optimizer=self.optimizer_dict[id(p)], **sched_kwargs)
>                 if accelerator is not None:
>                     self.scheduler_dict[id(p)] = accelerator.prepare_scheduler(self.scheduler_dict[id(p)])
>
>         if self.other_model is not None:
>             for p in self.other_model.parameters():
>                 if p.requires_grad:
>                     self.optimizer_dict[id(p)] = optimizer_class([{"params":p}], **optim_kwargs)
>                     if accelerator is not None:
>                         self.optimizer_dict[id(p)] = accelerator.prepare_optimizer(self.optimizer_dict[id(p)])
>
>             for p in self.other_model.parameters():
>                 if p.requires_grad:
>                     self.scheduler_dict[id(p)] = get_scheduler(optimizer=self.optimizer_dict[id(p)], **sched_kwargs)
>                     if accelerator is not None:
>                         self.scheduler_dict[id(p)] = accelerator.prepare_scheduler(self.scheduler_dict[id(p)])
>
>     def insert_hook(self, optimizer_class=None, get_scheduler=None, accelerator=None,
>                  optim_kwargs={}, sched_kwargs={}):
>
>         for param in self.model.parameters():
>             param.requires_grad = True
>
>         def optimizer_hook(p):
>             if p.grad is None:
>                 self.optimizer_dict[id(p)].zero_grad(set_to_none=True)
>                 self.optimizer_dict[id(p)].state_dict().clear()
>                 self.scheduler_dict[id(p)].state_dict().clear()
>                 del self.optimizer_dict[id(p)]
>                 del self.scheduler_dict[id(p)]
>                 return
>             else:
>                 if self.grad_aware:
>                     with torch.no_grad():
>                         self.grad_record[id(p)].append(((p ** 2).sum() / p.numel()))
>                         while len(self.grad_record[id(p)]) > 200:
>                             self.grad_record[id(p)].pop(0)
>
>                 if id(p) not in self.optimizer_dict:
>                     self.optimizer_dict[id(p)] = optimizer_class([{"params":p}], **optim_kwargs)
>                     if accelerator is not None:
>                         self.optimizer_dict[id(p)] = accelerator.prepare_optimizer(self.optimizer_dict[id(p)])
>                 if id(p) not in self.scheduler_dict:
>                     self.scheduler_dict[id(p)] = get_scheduler(optimizer=self.optimizer_dict[id(p)], **sched_kwargs)
>                     self.scheduler_dict[id(p)].last_epoch = self.last_epoch
>                     if accelerator is not None:
>                         self.scheduler_dict[id(p)] = accelerator.prepare_scheduler(self.scheduler_dict[id(p)])
>
>             if accelerator is not None and accelerator.sync_gradients:
>                 accelerator.scaler.unscale_(self.optimizer_dict[id(p)])
>                 torch.nn.utils.clip_grad_norm_(p, 1.0)
>
> ```

---

> ### Public Comment · ~Shitong_Shao1 · 2024-11-22
> **LISA and SaRA [3/3]**
>
> ```python
>             self.optimizer_dict[id(p)].step()
>             self.optimizer_dict[id(p)].zero_grad(set_to_none=True)
>             self.scheduler_dict[id(p)].step()
>
>         # Register the hook onto every parameter
>         for p in self.model.parameters():
>             if p.requires_grad:
>                 p.register_post_accumulate_grad_hook(optimizer_hook)
>
>         def spatial_optimizer_hook(p):
>             self.optimizer_dict[id(p)].step()
>             self.optimizer_dict[id(p)].zero_grad(set_to_none=True)
>             self.scheduler_dict[id(p)].step()
>
>         if self.other_model is not None:
>             for p in self.other_model.parameters():
>                 if p.requires_grad:
>                     p.register_post_accumulate_grad_hook(spatial_optimizer_hook)
>
>     @property
>     def get_better_parameters_name(self):
>         lisa_target_modules = [
>                 "to_q",
>                 "to_k",
>                 "to_v",
>                 "to_out.0",
>                 "proj_in",
>                 "proj_out",
>                 "ff.net.0.proj",
>                 "ff.net.2",
>                 "conv1",
>                 "conv2",
>                 "conv_shortcut",
>                 "downsamplers.0.conv",
>                 "upsamplers.0.conv",
>                 "time_emb_proj",
>             ]
>         return lisa_target_modules
> ```
>
> You can use it as follows
>
> ```python
>
> # Initialize your model
>
>     lisa_trainer = LISADiffusion(unet, spatial_head, rate=0.25)
>     lisa_trainer.insert_hook(optimizer_class=optimizer_class,
>                         get_scheduler=get_scheduler,
>                         accelerator=accelerator,
>                         optim_kwargs=dict(lr=args.learning_rate,
>                                           betas=(args.adam_beta1, args.adam_beta2),
>                                           weight_decay=args.adam_weight_decay,
>                                           eps=args.adam_epsilon),
>                         sched_kwargs=dict(name=args.lr_scheduler,
>                                           num_warmup_steps=args.lr_warmup_steps,
>                                           num_training_steps=args.max_train_steps))
>     lisa_trainer.register(optimizer_class=optimizer_class,
>                         get_scheduler=get_scheduler,
>                         accelerator=accelerator,
>                         optim_kwargs=dict(lr=args.learning_rate,
>                                           betas=(args.adam_beta1, args.adam_beta2),
>                                           weight_decay=args.adam_weight_decay,
>                                           eps=args.adam_epsilon),
>                         sched_kwargs=dict(name=args.lr_scheduler,
>                                           num_warmup_steps=args.lr_warmup_steps,
>                                           num_training_steps=args.max_train_steps))
> # ...
> global_step=0
> for i in range(epoch):
>   for image in enumerate(dataloader):
>     # forward
>     if global_step % 20 == 0 and global_step != 0: # you can use other number to replace 6
>       lisa_trainer.lisa_recall()
>       accelerator.clear()
>     global_step += 1
> ```

---

> ### Author Response · Authors · 2024-11-24
> **Response to Shitong Shao**
>
> Thanks for your attention.
>
> **1.** Your suggestion to improve the first and second-order moments in the optimizer is highly insightful, and we will further explore this in the future.
>
> **2.** Additionally, our method can indeed be applied to other approaches. Inspired by an analysis of the pre-trained diffusion model parameters, we developed our method to enhance the fine-tuning capabilities of diffusion models. In the future, we will explore its potential applications in other models.
>
> **3.**  Lastly, It seems that our method is not similar to LISA. LISA primarily focuses on a Layerwise Importance Sampling approach for LLMs, whereas our method introduces a progressive low-rank sparse fine-tuning approach tailored for diffusion models. The two methods are quite different in terms of both motivation and technical details. In the revised version, we have cited LISA to provide a more comprehensive discussion of the existing fine-tuning methods.

---

### Author Response · Authors · 2024-11-23
**Summary Of Changes**

We sincerely appreciate all Reviewers and the Area Chair for their time and efforts for our paper. Following the valuable suggestions and insights provided in the reviews, we have carefully revised the manuscript with changes highlighted in **blue**. Specifically, we have made several major changes in the following aspects:

1. We have compared our SaRA with more state-of-the-art methods, including **DoRA [1]**, and **DiffPruning [2]**, which are the representative RFT and SFT methods. The results further validate the effectiveness of our method. (Sec. C in Appendix)

2. We have conducted more comparison experiments on **Stable Diffusion XL 1.0 [3]**, which demonstrates the robustness of our method across different backbones. (Sec. C in Appendix)

3. We use an additional metric from **T2I-CompBench++ [4]** to evaluate the fine-grained alignment between the output image and the given text, where the results still show our model's superior ability to keep the model priors. (Sec. C in Appendix)

4. We have studied the **scale weight for SaRA parameters** and find that as the weight increases, the generated images will contain more task-specified features. (Sec. F in Appendix)
5. We add more studies on **merging SaRA parameters** learned from different datasets, enabling a more flexible application of our SaRA. (Sec. G in Appendix)
6. We conduct additional **ablation studies on the threshold** and find that SaRA with a bigger threshold can learn more task-specified knowledge. And with the help of our low-rank loss, SaRA will not overfit as the full-parameter fune-tuning methods. (Sec. H in Appendix)
7. We have added **more visualization results** of our method to show the generation diversity of our SaRA. (Sec. C in Appendix)
8. We have added a **limitation section** in the end of the Suppl to study the potential limitations of our methods, which gives a more comprehensive analysis of our SaRA. (Sec. M in Appendix)
9. We have corrected the mentioned typos and carefully checked our paper again. Moreover, we have also changed the figures as suggested.


**Reference**

[1] Liu S, Wang C Y, Yin H, et al. DoRA: Weight-Decomposed Low-Rank Adaptation[C]. ICML, 2024

[2] Guo D, Rush A M, Kim Y. Parameter-efficient transfer learning with diff pruning[J]. arXiv preprint arXiv, 2020.

[3] Podell D, English Z, Lacey K, et al. SDXL: Improving Latent Diffusion Models for High-Resolution Image Synthesis[C]. ICLR, 2024.

[4] Huang K, Sun K, Xie E, et al. T2i-compbench: A comprehensive benchmark for open-world compositional text-to-image generation[J]. NeurIPS, 2023.

Detailed point-by-point responses are listed for each **Reviewer** below.

---

### Author Response · Authors · 2024-11-25
**Please let us know whether we address all the issues (SaRA #229)**

Dear Reviewers,

Thank you for your valuable comments on our paper.

We have submitted our responses to your comments and included additional results in the revised version of the manuscript. Please let us know if you have any further questions or require additional clarifications so that we can address them during the discussion period. We hope that after we have addressed all the issues, you will consider raising the score of our submission.

Thank you for your time and consideration.

Best regards,

Authors of SaRA #229

---

### Comment · Area_Chair_WBbm · 2024-11-25
**Please check the authors' responses**

Dear reviewers,

Could you please check the authors' responses, and post your message for discussion or changed scores?

best,

AC

---

### Meta-Review · Area_Chair_WBbm · 2024-12-16

**Metareview:**

This paper works on the fine-tuning of diffusion models for new tasks. The basic idea is to identify the importance of parameters in the pre-trained diffusion model as the parameters with smallest absolute values. Then the proposed method fine-tunes these ineffective parameters using sparse weight matrix, with nuclear-norm-based low-rank regularization in fine-tuning. The proposed approach was applied to the downstream fine-tuning tasks on five styles, showing the better performance in FID, CLIP and VLHI metrics, compared with the LoRA, Adaptformer, LT-SFT.  This proposed approach tackles an important task for fine-tuning the diffusion models based on parameter selection and optimization. The reviewers commented on the strength in the experimental results and comparisons, and the fundamental idea of storing certain gradients and memory-efficiency, etc.

**Additional Comments On Reviewer Discussion:**

Reviewer RosU raised questions on comparison with sota PEFT techniques,  and working on stable diffusion XL 1.0 version, etc. Reviewer EfNM raised concerns on the simple reuse of unimportant parameters, and how to merge multi-tasks’ parameter and the performance. Reviewer GkU7 is more positive and rated score of 8. The major questions are on the results with same prompts but different seeds, and more visual results.  Reviewer bh95 concerns on the novelty of the selecting parameters, lacking comparison with other SFT methods, performance improvements, and unclear reason of inefficient parameters becoming efficient in training.  Reviewer GTZk concerns on the applications across different networks or frameworks, effect of the choice of thresholds for low-absolute-value parameters, and analysis of limitations.

In the discussion phase, Reviewer RosU raised the scores to 6, while Reviewer EfNM did not update decision in the post-rebuttal phases.  Reviewer bh95 thinks that most of the concerns have been addressed and more comparisons with STF methods were further provided by authors. Reviewer GTZk was also satisfied with the responses and raised the score.

Though Reviewer EfNM did not provide updated decision, the reviewer's concerns have been addressed in the authors' responses. Considering most positive final decisions of these reviewers, the paper can be accepted.

---

### Decision · Program_Chairs · 2025-01-22

Accept (Poster)